# Health-related quality of life is linked to the gut microbiome in kidney transplant recipients

J. Casper Swarte [1,2,20], Tim J. Knobbe [2,20], Johannes R. Björk [1,3], Ranko Gacesa [1,3], Lianne M. Nieuwenhuis[1], Shuyan Zhang [1], Arnau Vich Vila [1,3], Daan Kremer [2], Rianne M. Douwes[1,2], Adrian Post[2], Evelien E. Quint[4], Robert A. Pol[4], Bernadien H. Jansen[1], TransplantLines investigators*, Martin H. de Borst [2], Vincent E. de Meijer [5], Hans Blokzijl[1], Stefan P. Berger [2], Eleonora A. M. Festen[1,3], Alexandra Zhernakova[3], Jingyuan Fu[3,6], Hermie J. M. Harmsen [7], Stephan J. L. Bakker [2,21] & Rinse K. Weersma [1,21] ✉

Kidney transplant recipients (KTR) have impaired health-related quality of life (HRQoL) and suffer from intestinal dysbiosis. Increasing evidence shows that gut health and HRQoL are tightly related in the general population. Here, we investigate the association between the gut microbiome and HRQoL in KTR, using metagenomic sequencing data from fecal samples collected from 507 KTR. Multiple bacterial species are associated with lower HRQoL, many of which have previously been associated with adverse health conditions. Gut microbiome distance to the general population is highest among KTR with an impaired physical HRQoL ($R = -0.20$, $P = 2.3 \times 10^{-65}$) and mental HRQoL ($R = -0.14$, $P = 1.3 \times 10^{-3}$). Physical and mental HRQoL explain a significant part of variance in the gut microbiome ($R^2 = 0.58\%$, FDR $= 5.43 \times 10^{-4}$ and $R^2 = 0.37\%$, FDR $= 1.38 \times 10^{-3}$, respectively). Additionally, multiple metabolic and neuroactive pathways (gut brain modules) are associated with lower HRQoL. While the observational design of our study does not allow us to analyze causality, we provide a comprehensive overview of the associations between the gut microbiome and HRQoL while controlling for confounders.

Kidney transplantation is the preferred treatment of patients with end-stage kidney disease and improves survival after transplantation compared with patients who are treated with dialysis[1,2]. However, health-related quality of life (HRQoL) of kidney transplant recipients (KTR) still remains lower after transplantation compared with the general population, especially regarding physical HRQoL[2]. Improving HRQoL in the long term after transplantation would greatly improve the outcomes of kidney transplantation.

The gut-brain axis refers to the bidirectional communication between the gut and the brain, which plays a role in regulating mood,

behavior, and overall well-being. The gut and the central nervous system are known to communicate via neural, immunological and chemical pathways[3]. Therefore, it is not surprising that gut health and HRQoL are tightly connected[4]. The gut microbiome can influence the central nervous system via the gut-brain axis[5,6] with, for example, bacterial cell wall components[7] or short chain fatty acids[8]. Translation of these mostly animal-based studies to human subjects remains difficult, although it has previously been shown that modifying dietary fiber intake is associated with improved mental HRQoL[9], which could be mediated by the gut microbiome[10].

---

A full list of affiliations appears at the end of the paper. *A list of authors and their affiliations appears at the end of the paper. ✉ e-mail: r.k.weersma@umcg.nl

Recently, a cross-sectional study in the general population identified the gut microbiome as a factor associated with HRQoL[11]. Multiple microbial genera and their neuroactive functions appear to be associated with domains of HRQoL in the general population[11]. However, it is currently unknown if these associations are also present in KTR. Previous studies have also shown that KTR suffer from gut dysbiosis which was associated with increased mortality after transplantation[12,13].

In this study, we aimed to identify any relationship between gut microbiome dysbiosis and impaired HRQoL among KTR, using shotgun metagenomic sequencing data. This method allows to analyze gut microbial composition, metabolic function and neuroactive metabolic modules, and to link those to both physical and mental components of HRQoL assessed from 507 KTR part of the TransplantLines Biobank and Cohort study[14]. We included cross-sectional data of KTR who were at least 1 year after transplantation to reflect the population of KTR at medium- to long-term after transplantation. Understanding the link between the gut microbiome and HRQoL could help improve the quality of life of KTR and likely also other solid organ transplant recipients.

## Results

### HRQoL of kidney transplant recipients

Of the 751 KTR that provided a fecal sample in the TransplantLines Biobank and Cohort study, HRQoL data were available for 507 (68%) recipients. The average age was $57 \pm 13$ years, and 45% of the recipients were female. The median time after transplantation was 5.0 years [IQR 1.0-12.0] (Supplementary Fig. 1), average estimated glomerular filtration rate (eGFR) at inclusion was $53 \pm 18$ mL/min/1.73m$^2$, and 71% of KTR were dependent on dialysis before transplantation. Most KTR (46%) used triple immunosuppressive therapy consisting of prednisolone, tacrolimus and mycophenolate mofetil, 19% KTR were on a different triple immunosuppressive therapy consisting of prednisolone, a calcineurin inhibitor and a proliferation inhibitor, and 35% were on a double immunosuppressive therapy. The average physical HRQoL (physical component score) was $68.7 \pm 22.2$ (range: 3.8–100.0) and the average mental HRQoL (mental component score) was $76.4 \pm 17.7$ (range: 15.9–100.0). More extensive characteristics, results of physical assessments and patient reported outcome measures are presented in Supplementary Data 1. We analyzed the association between HRQoL components and phenotypes that have previously been shown to potentially confound gut microbiome analyses[15], and accounted for these covariates (age, body mass index, sex, stool water content, diabetes, dialysis-dependency before transplantation, anti-hypertensive treatment, proton pump inhibitors, laxatives and antibiotics) in all downstream analyses (Supplementary Data 2).We did not have the Bristol stool scale available for all KTR, however, we measured stool water content which is a proxy for stool consistency and transit time[16]. For comparison 151 healthy controls who provided a fecal sample and responded to the SF-36 questionnaire were included that participated in the TransplantLines study[14].

### Variation in HRQoL is associated with gut microbial composition in kidney transplant recipients

The SF-36 assesses HRQoL using 8 domains: general health, physical health, role limitations due to impairment of physical health, pain, emotional well-being, role limitations due to emotional problems, impaired social functioning and impaired vitality[17]. These scores are summarized in the physical and mental component score (hereafter PCS and MCS, respectively) reflecting physical and mental HRQoL. Each domain has a possible range between 0 and 100 with a score of 0 representing a perceived worse health and a score of 100 representing a perceived perfect health. KTR score lower on all HRQoL features compared with 151 healthy controls from the TransplantLines study (Wilcoxon, $P < 1.43 \times 10^{-7}$; Supplementary Fig. 2). While PCS and MCS capture different domains of HRQoL they are strongly correlated

($r = 0,69$, Spearman, $P = 1.08 \times 10^{-77}$). We found that the mean standardized PCS score (standardized based on the mean and standard deviation of the United States general population)[18] of our KTR population was 44.5 (SD 10.7), and the mean standardized MCS score was 52.8 (SD 8.6). Using the Dutch population to standardize the PCS and MCS scores, mean standardized PCS and MCS scores were 45.2 (SD 10.2) and 50.6 (SD 8.9), respectively. As a score of 50 is regarded as the mean HRQoL-score of the general population[18], this implies that the HRQoL of KTR is on average lower in PCS and higher in MCS than the general population. In total 303 (60%) of KTR had a score <50 on the PCS and 137 (27%) of KTR had a score <50 on the MCS. Thus, there is a large variation in how population experiences physical HRQoL (PCS range: 3.8–100.0) and mental HRQoL (MCS range: 15.9–100.0). We captured this variation by discretizing both scores into quartiles for depiction (Fig. 1a, b; PCS: Q1[3.8–50.0]; Q2[50.0–75.6]; Q3[75.6–87.5]; Q4[87.5–100.0] and MCS: Q1[15.9–68.0]; Q2[68.0–81.9]; Q3[81.9–89.0]; Q4[89.0–100]).

To assess the relationship between gut microbial composition and HRQoL, we analyzed beta diversity and performed principal component analysis (PCA) on clr-transformed relative abundances which produces the Aitchison distance (the gold standard for compositional data[19]). PCA is a dimension reduction technique for high dimensional data (e.g. bacterial species in the present analysis)[20]. We observed significant associations between PC1 and all HRQoL-domains and the summary scores (PCS and MCS) of these domains (Spearman correlation on continuous HRQoL scores, $P < 0.05$; Supplementary Data 3). Figures 1a, b reveals an interesting pattern: an association between PC1 and the quartiles of the component scores of HRQoL. Consequently, we found that PC1 was significantly different between the lowest (Q1) and the highest quartile (Q4) of each component scores of HRQoL (PCS: $P = 7.2 \times 10^{-6}$; MCS: $P = 1.2 \times 10^{-2}$; Fig. 1c, d).

A previous study showed that all domains of HRQoL are associated with interindividual variation in gut microbiome composition of a general population[11]. To analyze whether HRQoL explained interindividual variation in the gut microbiome in our KTR population, we performed Permutational Multivariate Analysis of Variance (PERMANOVA) on Aitchison distances. Similarly, to the previous study by Valles-Colomer et al., we found that all domains of HRQoL, apart from the emotional wellbeing domain, explained variation in the gut microbiome of KTR that was statistically significant (FDR < 0.05; Fig. 2, Supplementary Data 4). We found that the domain physical functioning explained the most variance ($R^2 = 0.63$%, FDR = $5.43 \times 10^{-4}$) followed by PCS ($R^2 = 0.58$%, FDR = $5.43 \times 10^{-4}$) and MCS ($R^2 = 0.37$%, FDR = $1.38 \times 10^{-3}$). In comparison to the variance explained by factors such as proton pump inhibitor (PPI) use and age (PPI use: $R^2 = 1.10$%, FDR = $1.00 \times 10^{-4}$; age: $R^2 = 0.56$%, FDR = $1.00 \times 10^{-4}$), HRQoL explained a moderate amount (Fig. 2, Supplementary Data 4).

### Previously identified disease-associated species are associated with a lower HRQoL

The Dutch Microbiome Project—a characterization of the gut microbiome of 8208 individuals from the northern Netherlands—revealed that many bacterial species are associated with self-reported disease[21]. The 507 KTR in the current study[14] are from the same geographical region but were not part of the Dutch Microbiome Project. To further characterize the observed differences in gut microbial composition (as mainly characterized by PC1) between the quartiles of the component scores of HRQoL (PCS and MCS; Fig. 1a, b), we compared bacterial species that loaded strongly onto PC1 with bacterial species that were associated with disease status in the Dutch Microbiome Project. Interestingly, we observed a striking similarity with species driving the variation in PC1 exhibited negative correlations with disease-associated species, and positive correlations with species not associated with disease[21] (Fig. 1e). This suggests that similar species that are associated with disease in the general population are associated with

lower HRQoL in KTR. A complete list of the correlation between species loading onto different PCs and HRQoL features can be found in Supplementary Data 3.

To further analyze these findings, we quantified gut microbial dysbiosis of each KTR. We use the same definition of gut microbial dysbiosis that we previously found to be predictive of mortality after transplantation[13], which is defined as the microbiome distance

between each KTR to the average of the general population. We investigated whether the extent of dysbiosis is associated with lower HRQoL, and indeed found that the distance between each KTR and the general population was significantly associated with physical HRQoL (PCS: $r = -0.20$, $P = 2.3 \times 10^{-6}$), mental HRQoL (MCS: $r = -0.14$, $P = 1.0 \times 10^{-3}$) and all eight domains of HRQoL ($r < -0.10$, $P < 0.01$), apart from the mental health domain (Supplementary Data 5, Fig. 1c,

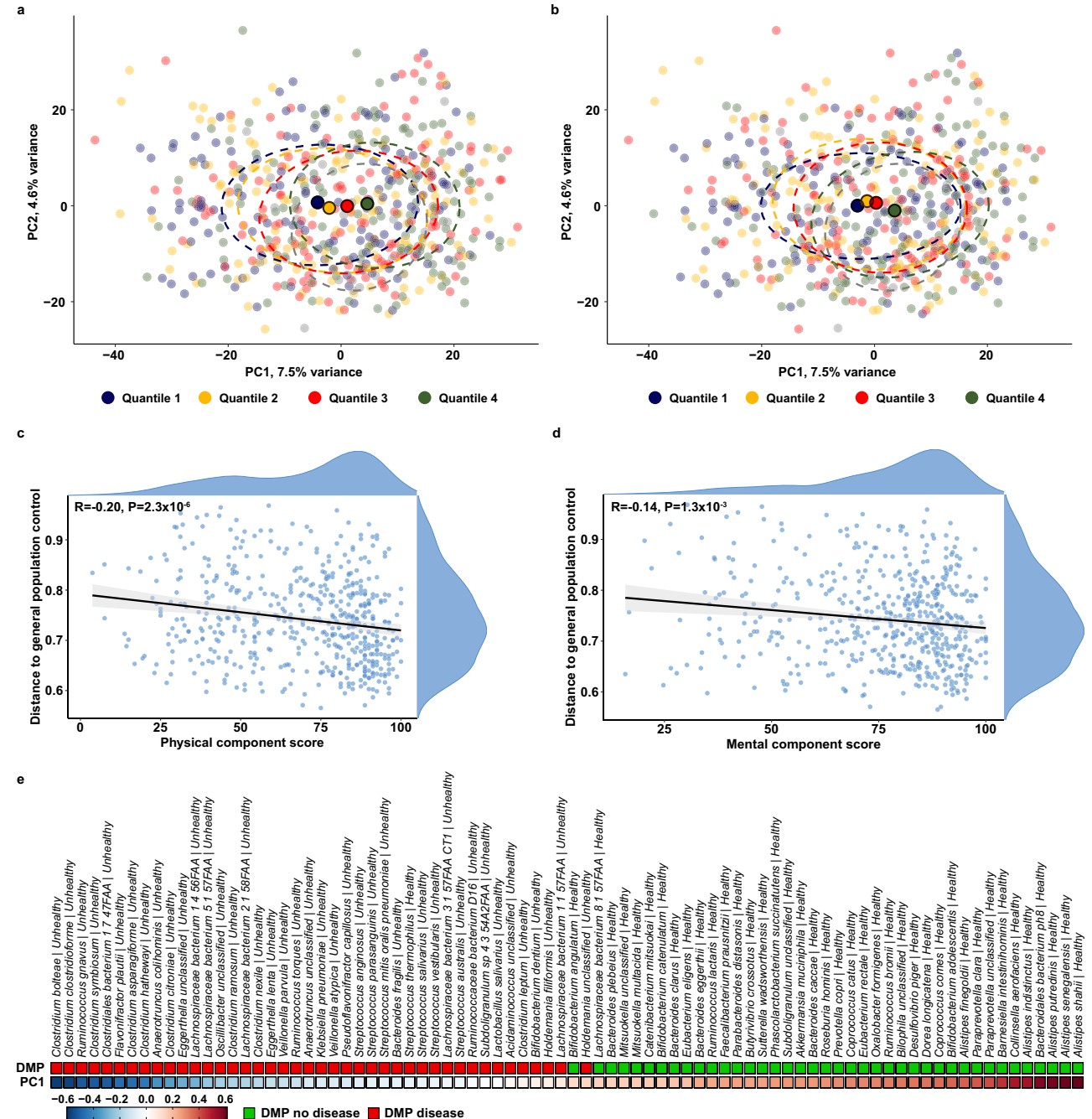

**Fig. 1 | Disease-associated bacterial species in the gut microbiome of KTR are associated with HRQoL. a, b** Principal component analysis on the clr-transformed species reflecting the Aitchison distances between KTR. The physical component score (**a**) and the mental component score (**b**) are divided into quartiles (KTR were divided into quartiles (Q1, Q2, Q3, Q4) based on their PCS and MCS with Q1 containing the lowest HRQoL scores and Q4 the highest HRQoL scores). The large dots represent centroids per group and the dashed circles represent 95% confidence ellipses. **c, d** Correlation plot with spearman correlation and 95% confidence interval for the physical component score (**c**) the mental component score (**d**) and

the distance to general population controls. This dysbiosis score was calculated previously by calculating the Aitchison distance from KTR with 1183 age-, sex- and BMI-matched general population controls[13]. **e** Heatmap depicting significant correlations between species that have previously been associated with disease vs. no disease in the Dutch microbiome project (DMP)[21]. Species that are associated with no disease in the DMP (green squares) were consistently, positively and significantly associated with principal component 1 (i.e. higher HRQoL) in our study while the opposite effect was observed for species that were associated with disease in the DMP.

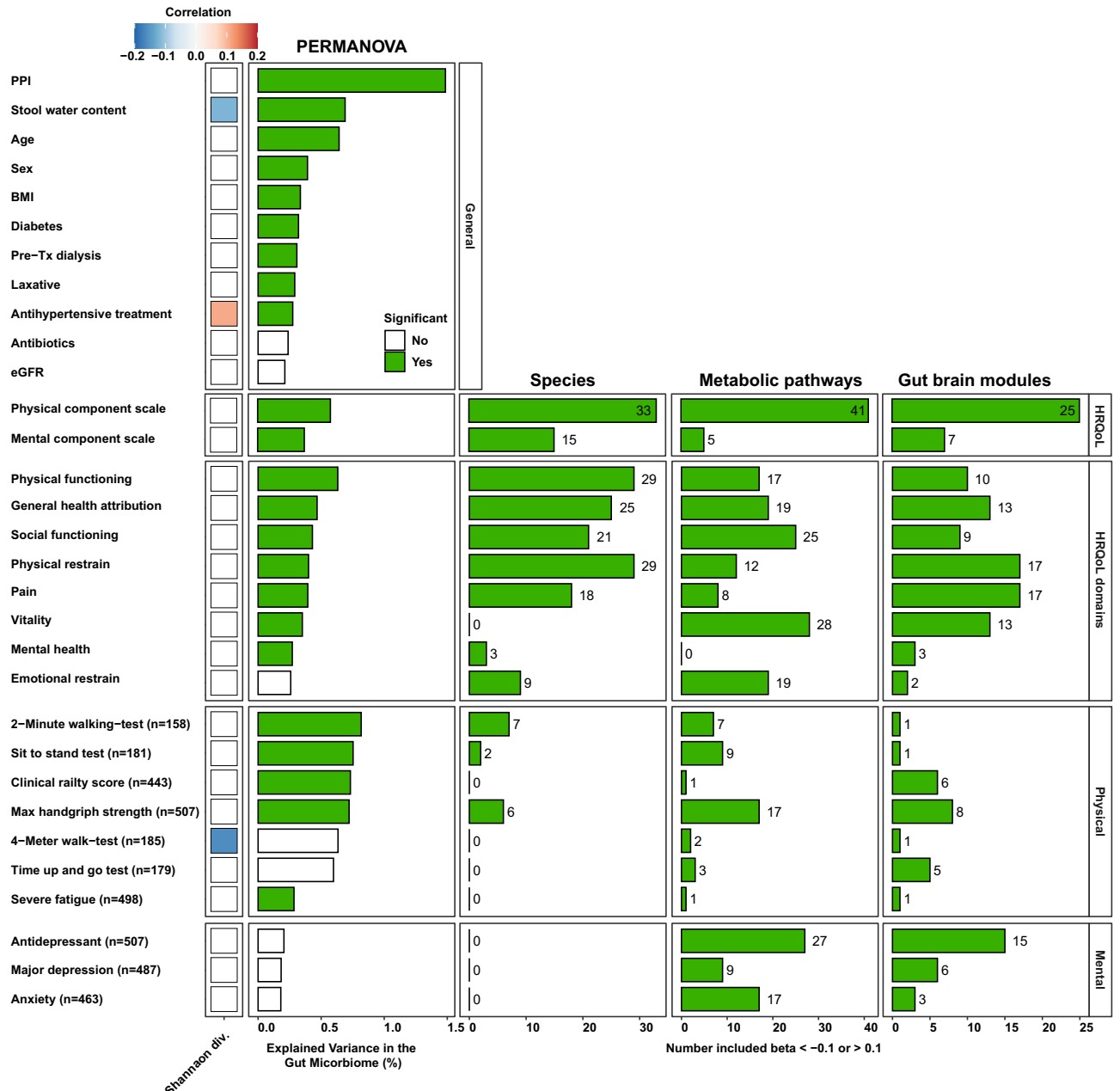

**Fig. 2 | Overview of associations of HRQoL and physical and mental assessments with gut microbial species.** Heatmap depicting significant spearman correlations (indicated with blue color for negative correlations and red for positive correlations) of general, HRQoL, physical health and mental health variables with Shannon diversity index (Shannon div.). In the Permutational Multivariate Analysis of Variance (PERMANOVA) column the percentage of explained variance from the ADONIS analysis are shown. Bars in green represent the number of included microbiome features in the elastic-model with a **β** < -0.1 or (**a**) **β** > 0.1. In the columns, species, metabolic pathways and gut brain modules, the number of selected features for the elastic net model are depicted. Shannon div.: Shannon diversity, eGFR: estimated glomerular filtration rate, Pre-Tx: pre-transplantation, FDR: false discovery rate.

d). The distance to our general population was highest in the first (Q1) compared to the last quartile (Q4) of the HRQoL component scores (PCS: $P = 4.9 \times 10^{-6}$; MCS: $P = 7.0 \times 10^{-2}$). We next quantified species-level alpha diversity. However, we did not find any significant associations between species richness, Shannon diversity, or the Simpson index and the different HRQoL domains ($P > 0.05$; Supplementary Fig. 3).

**Multiple gut microbial species are associated with HRQoL**
To test the relationship between HRQoL and the gut microbiome, we used elastic net regularization, an approach that performs both variable selection and regularization[22]. This technique is favorable when

you have a large number of independent variables (in our case bacterial species, metabolic pathways or gut brain modules) that are strongly correlated[23–25]. We used a 10-fold cross-validation approach with a 75–25% train-test split (see Methods) to test how well our models generalized on "untouched" data. In the following text, we are only reporting microbial features whose coefficients (**β**'s) were selected by the model (i.e. those with an effect size larger than zero). We included all potential confounder variables that we described previously in the elastic net model. The same analysis was performed in a cohort of 151 healthy controls to disentangle a KTR-specific signal.

The model for physical HRQoL (PCS) included the highest number of bacterial species 33 (16%) while the model for mental HRQoL (MCS)

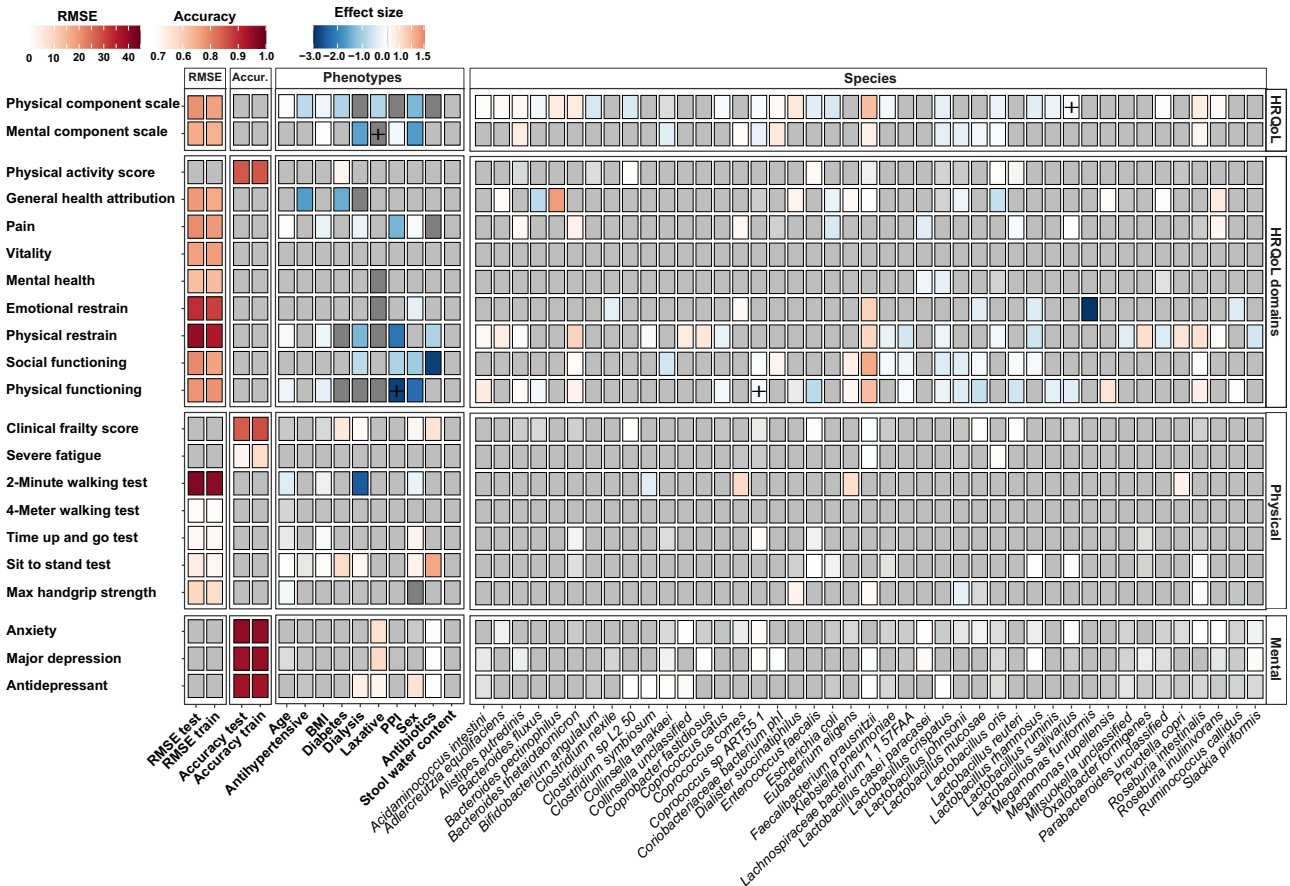

**Fig. 3 | Multiple bacterial species in the post-transplant gut microbiome are associated with HRQoL and assessments reflecting physical and mental health.** This heatmap depicts the results from the elastic net analysis for associations with a $\beta > 0.3$ or $\beta < 0.3$ that was performed for HRQoL scores and assessments of physical and mental health. For numerical variables, the RMSE of each elastic net model is shown in the first two columns (test and train set, respectively) and for categorical variables, the accuracy is depicted in the next two columns (test and train set, respectively). Selected features in the elastic net model are colored by effect size. Gray tiles encompass phenotypes or species that were not included in the model. Plus-sign depict associations that were also observed in the control cohort. RMSE: Root-mean-square error, Accur.: accuracy.

included 15 (7%) bacterial species, as presented in Fig. 2. In total, 14 (7%) species were included in both the physical and mental HRQoL models. The strongest and most consistent signals were found for *Faecalibacterium prausnitzii*, which was associated with higher physical ($\beta = 0.93$) and mental ($\beta = 0.27$) HRQoL (Fig. 3, Supplementary Data 6), similar to what has previously been observed in the general population[11]. *F. prausnitzii* is one of the most common gut bacteria and tends to be less abundant in the gut microbiome of individuals with disease[21]. Multiple bacteria were associated with physical HRQoL, of which *Dialister succinatiphilus* ($\beta = 0.39$), *Bacteroides thetaiotaomicron* ($\beta = 0.34$), *Roseburia intestinalis* ($\beta = 0.34$) and *Bacteroides pectinophilus* ($\beta = 0.33$) were most strongly positively associated, and *Bifidobacterium angulatum* ($\beta = -0.53$), *Escherichia coli* ($\beta = -0.53$), *Clostridium sp_L2_50* ($\beta = -0.49$), *Enterococcus faecalis* ($\beta = -0.40$), *Lactobacillus rhamnosus* ($\beta = -0.40$) and *Lactobacillus crispatus* ($\beta = -0.36$) were most strongly negatively associated with physical HRQoL (Fig. 3, Supplementary Data 6). Interestingly, multiple *Bacteroides* ($n = 4$), *Bifidobacterium* ($n = 4$), *Clostridium* ($n = 5$) and *Lactococcus* ($n = 6$) *species* were less abundant in the gut microbiome of KTR with a lower physical HRQoL, and *Klebsiella pneumoniae* ($\beta = -0.17$) and *Streptococcus parasanguinis* ($\beta = -0.09$) were more abundant in KTR with a lower physical HRQoL (Supplementary Data 6). We found that the same 14 (7%) species were included in both the model for physical HRQoL and mental HRQoL. However, 24 species included in the mental HRQoL model were not included in the physical HRQoL model, including; *Lactobacillus johnsonii* ($\beta = -0.33$), *Odoribacter unclassified*

($\beta = -0.24$), *Bifidobacterium longum* ($\beta = 0.18$) and *Anaerostipes hadrus* ($\beta = 0.14$; Fig. 3, Supplementary Data 6). Most of the observed associations between HRQoL and the gut microbiome appear to be specific to KTR as many of the associations were not observed in the control cohort (Fig. 3, Supplementary Data 6). To contrast these results, we also performed an association analysis with a simple linear regression model per species. For this analysis, we constructed three models: (i) a model without confounders; (ii) a model with microbiome associated covariates (i.e. stool water content, antibiotics, laxatives, proton pump inhibitors); and finally (iii) a model including all confounders (age, sex, BMI, eGFR, stool water content, antibiotics, antihypertensive treatment, diabetes, dialysis, laxative, proton pump inhibitors). This analysis revealed many significant associations between HRQoL features and gut microbial species, many of which were also included in the elastic net model. For example, similarly to the elastic net, this analysis also found a significant positive association between *F. prausnitzii*, *R. intestinalis* and *Bacteroides thetaiotaomicron* and physical HRQoL. Similarly, *Clostridium clostridioforme* was also found by this analysis to be associated with a decreased physical HRQoL (Supplementary Data 7). However, comparing the results from the elastic net and this analysis also revealed many differences, which are to be expected. Importantly, compared to simple linear regression, elastic net incorporates L1 and L2 regularization which means that a large number of correlated predictor variables can be modeled while minimizing overfitting and reducing bias due to multicollinearity[26].

Next, we analyzed which metabolic pathways were associated with physical and mental HRQoL. In total, 41 (12%) pathways were included in the model for physical HRQoL and 5 (2%) pathways were included in the model for mental HRQoL (Fig. 2), 4 (1%) of these pathways were included in both the model for physical and mental HRQoL. For physical HRQoL, the strongest positive signals we observed were for inosine biosynthesis (PWY6124, $\beta = 1.65$), rhamnose biosynthesis ($\beta = 1.41$), stachyose degradation (PWY6527, $\beta = 0.90$), pyruvate fermentation pathway (PWY7111, $\beta = -0.52$). We also observed 10 quinone biosynthesis pathways which were lower in KTR with a lower physical HRQoL (Fig. 3, Supplementary Data 8). Interestingly, KTR with lower physical HRQoL also had a lower abundance of butyrate producing pathways (PWY5676, $\beta = -0.19$). We found five metabolic pathways that were associated with physical HRQoL which were also observed in the general population (Supplementary Data 8). No other pathway related associations were observed in our control population. In Supplementary Data 9 we report per species associations between all tested phenotypes and the aforementioned three simple linear regression models.

## The gut microbial neuroactive potential is associated with HRQoL

To analyze the relationship between the central nervous system and the gut microbiome beyond taxonomic associations, Valles-Colomer et al. developed the omixerRPM package[11]. This framework reclassifies KEGG orthologs[27] into gut-brain modules (GBM) which are microbial pathways that metabolize molecules that potentially interact with the human nervous system[11]. Because self-reported HRQoL is a subjective measure, we hypothesized that neuroactive compounds in the gut can potentially influence an individual's perceived HRQoL. This hypothesis is partly driven by the observation of a relationship between the neuroactive potential and HRQoL in the general population[11]. We applied this framework on our metagenomic sequencing data and found 56 gut-brain modules (GBMs) which primarily correspond to neuroactive compound production and degradation[11]. We applied the same elastic net approach, as described above, to test the relationship between HRQoL and GBMs. In total, 25 (45%) GBMs were selected by the model with physical HRQoL (PCS) and 7 (13%) GBMs were selected by the model with mental HRQoL (MCS; Fig. 2), with 10 (18%) GBMs being shared across both models. KTR with a higher physical HRQoL had less abundant pathways for isovaleric acid synthesis I (GBM034, $\beta = -1.35$), less menaquinone synthesis (GBM041, $\beta = -0.97$), less GABA synthesis (GBM022, $\beta = -0.71$) and butyrate synthesis (GBM053, $\beta = -0.47$) while having higher pathways abundance for acetate synthesis (GBM043, $\beta = 5.86$), β-estradiol degradation (GBM031, $\beta = 1.46$) and glutamate synthesis (GBM007, $\beta = 0.85$) (Fig. 3, Supplementary Data 10). KTR with a better mental HRQoL had less abundant pathways for menaquinone synthesis (GBM041, $\beta = -1.48$), quinolinic acid synthesis (GBM032, $\beta = -1.06$) and inositol synthesis (GBM037, $\beta = -0.38$), while having more abundant pathways for acetate synthesis (GBM043, $\beta = 1.03$), glutamate synthesis (GBM007, $\beta = 0.89$; Fig. 3, Supplementary Data 10). Associations with multiple species in the post-transplant microbiome and HRQoL are presented in Fig. 3, in which we also present associations with physical and mental health. The results we observed in KTR were highly consistent with the results found by Valles-Colomer et al[11]. in the general population suggesting that neuroactive compounds which could potentially be produced by gut bacteria play a role in KTR's perceived HRQoL. Two GBM that were associated with vitality and seven GBM that were associated with general health attribution were also observed in our general population cohort (Supplementary Data 10). No other pathway related associations were observed in our general population. In Supplementary Data 11 we report per species associations between all tested phenotypes and the aforementioned simple linear regression models.

## Other physical and mental health related phenotypes and the gut microbiome

We next aimed to support the associations we found between the gut microbiome and HRQoL using multiple assessments of physical and mental health which are also available in the TransplantLines Biobank and Cohort study[14]. Using these more objective assessments, we were able to further support some of the associations between HRQoL and the gut microbiome we report above. Assessments reflecting physical health were: physical activity per week; clinical frailty (clinical frailty scale); feeling of severe fatigue (CIS20R questionnaire); 2-min walk test; 4-meter walk test; timed up and go test; sit to stand test; and handgrip strength[14]. Assessments reflecting mental health were: use of antidepressants; feeling of anxiety (STAI−6 questionnaire); and severe depressive symptoms (PHQ9 questionnaire). Although data regarding HRQoL were available in all 507 KTR, the additional physical and mental health assessments were available in a lower number of KTR (Supplementary Data 1).

We found that KTR who were clinically well (based on the frailty scale) scored higher on PC1 and had a lower microbiome distance to general population (1183 matched controls[13], see Methods) compared with clinically vulnerable (Wilcoxon, $P = 0.02$ and $P = 1.60 \times 10^{-2}$, respectively) and clinically frail (Wilcoxon, $P = 0.02$ and $P = 4.60 \times 10^{-2}$, respectively) KTR (Fig. 4a and c). Handgrip strength positively correlated with PC1 and negatively with the microbiome distance to general populations (Fig. 4D). KTR who performed worse on the 4-meter walk test had a lower Shannon diversity (Spearman, $R = -0.08$, $P = 0.02$, Fig. 4d). Furthermore, microbiome distance to the general population was lower for KTR who were less physically active or reported severe fatigue (Wilcoxon, $P < 2.20 \times 10^{-16}$). KTR with severe symptoms of depression, who used antidepressants or who suffered from anxiety exhibited lower Shannon diversity and higher microbiome distances to the general population ($P < 2.20 \times 10^{-16}$, Fig. 4b, c). While this analysis further supports some of our findings regarding the association between HRQoL and the microbiome distance to general population, including Shannon diversity, the findings of this analysis were not highly consistent with the differential abundance analysis (Fig. 3).

## Discussion

The current study provides strong support for an association between the gut microbiome and HRQoL in a large population of KTR. We found that gut microbial composition and HRQoL are associated in a similar manner as previously observed in the general population[11]. In addition, we found that bacterial species which were associated with lower HRQoL have previously been associated with disease in the Dutch microbiome project[21]. Moreover, the average microbiome distance to our general population was higher among KTR with an impaired HRQoL. We identified 33 bacterial species associated with physical HRQoL and 15 associated with mental HRQoL. Other associations regarding measures of physical and mental functioning support some of our reported associations between the gut microbiome and HRQoL among KTR.

Several studies have shown associations between the gut microbiome and HRQoL in other populations[9,11,28–32]. In this study we show that this association is also present among KTR. This is particularly important in this population given that many KTR suffer from gut dysbiosis[12,13,33] and because their HRQoL is generally impaired[2]. Interestingly, our results are in line with results found in the general population and in patients with depression[11]. Similar to observations in the general population[11], we found that the presence of F. prausnitzii was most strongly and consistently associated with higher HRQoL. F. prausnitzii is one of the most common gut bacteria and tends to be less abundant in the gut microbiome of individuals with disease[21]. Other bacteria that were associated with higher HRQoL were Dialiser succinatiphilus (physical HRQoL) and Coprococcus comes (mental HRQoL), which are also in agreement with the observations from the general

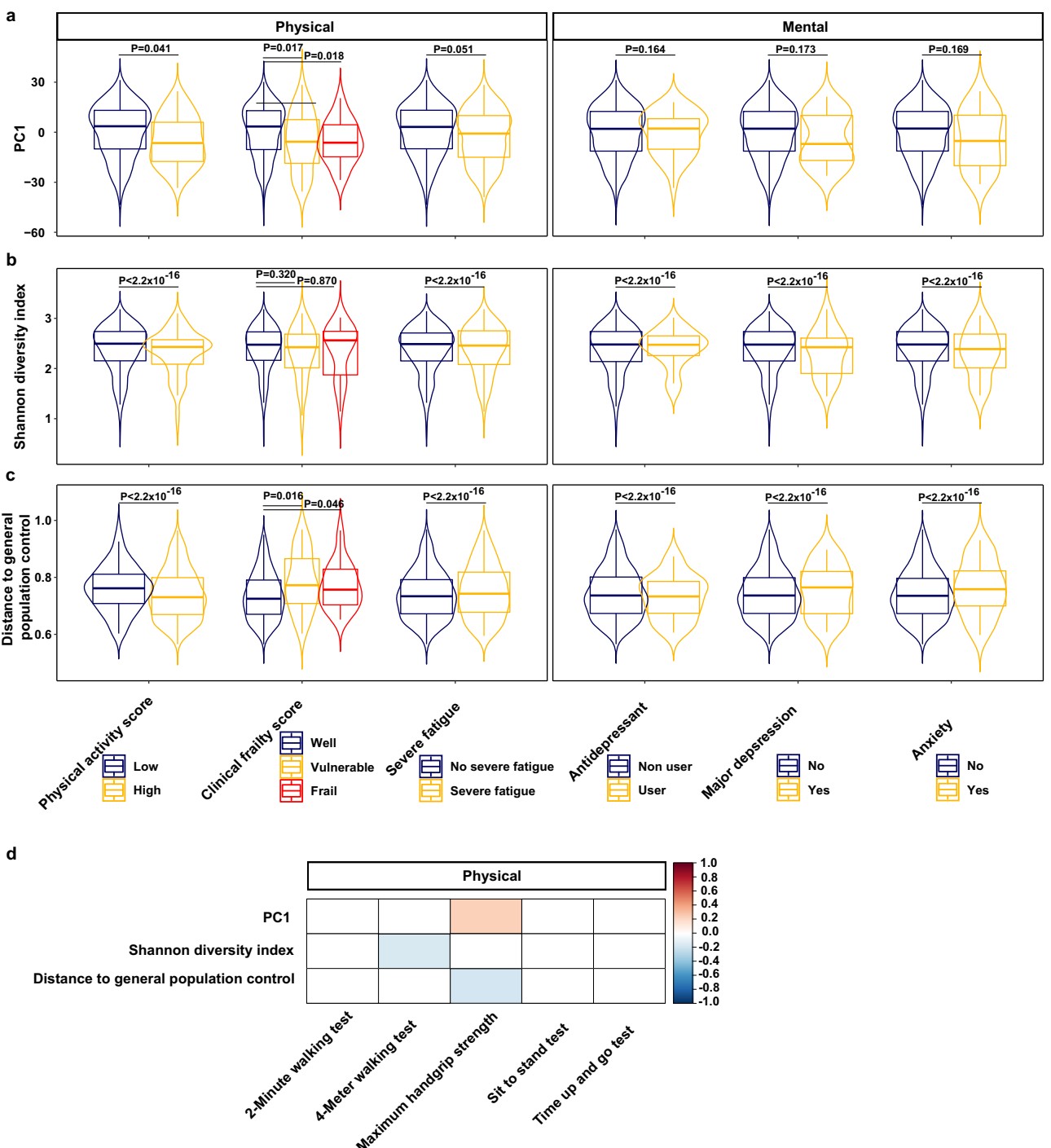

**Fig. 4 | Multiple assessments of physical- and mental health are associated with alpha- and beta diversity indices. a, b, c** Overview of PC1, Shannon diversity index and distance to general population controls for categorical physical- and mental phenotypes, $n = 507$. Boxplots depict minimal, quartiles, median and maximal values. A Wilcoxon rank sum test was used to test for significance. (**d**) Overview of PC1, Shannon diversity index and distance to general population controls for continuous physical phenotypes. We did not find any significant associations for the mental phenotypes using the spearman correlation test ($P > 0.05$).

population[11]. Notably, no association was found between any *Butyrivibrio spp.* and mental HRQoL, which was observed in the general population. Alongside *F. prausnitzii*, we found more butyrate producing bacteria to be positively associated with HRQoL: *Roseburia hominis, Alistipes putredinis, Eubacterium hallii, R. intestinalis* and *R. inulinivorans*[34]. Butyrate has multiple beneficial effects on human health, especially gut health by regulatory role in transepithelial fluid transport, ameliorating mucosal inflammation and oxidative stress and reinforcing the epithelial defense barrier[35]. Although we did not

measure fecal butyrate levels, these results suggest that gut microbial butyrate production could play a role in HRQoL in KTR. Multiple studies with participants from multiple populations, among which patients suffering from diabetes or chronic kidney disease, and patients undergoing hemodialysis or peritoneal dialysis and the general population[9,28,29,31,32,36], report a positive effect on HRQoL after administration of a pre- and/or probiotics. Given the association between the gut microbiome and HRQoL among KTR found in this study, we hypothesize that pre- and/or probiotics may also have

positive effects on HRQoL among KTR. Future interventional studies are warranted to confirm the potential benefit of pre- and/or probiotics on HRQoL among KTR. Interestingly we found that patients who are less physically active or who reported severe fatigue have a larger distance to the general population in terms of their gut microbiome composition. This could potentially be a reflection of poor general health and dysbiosis as a consequence of polypharmacy, antibiotics use and comorbidities[37]. We found multiple gut brain modules that were associated with HRQoL which suggests that gut microbial products could potentially play a role in perceived HRQoL via the gut-brain axis. However, it was rather surprising that only one gut brain module was uniquely associated with the MCS. This could potentially be attributed to the strong correlation between the PCS and MCS.

An important limitation of the current study is that we cannot infer causality or directionality in the reported association between the gut microbiome and HRQoL. However, it provides an overview of associations in the largest cross-sectional cohort of KTR with metagenomic sequencing data. Another limitation of the current study was the lack of support for more objective phenotypes regarding physical and mental well-being that were collected in the TransplantLines Biobank and Cohort study. Mental health was only assessed through self-report and not by health-care professional assessed diagnosis. Most of these phenotypes were only available in half of the KTR due to the design of the TransplantLines Biobank and Cohort study[14] which limits the interpretability of these associations. Data regarding diet was not available for all KTR in the current study and therefore, we could not disentangle the effect of diet in the observed associations between the gut microbiome and HRQoL. Future studies should include diet to further study these associations. We used elastic net analysis to accommodate for the high number of covariates (HRQoL features, other physical and mental health related phenotypes, confounders, bacterial species, metabolic pathways and gut brain modules) and account for the collinearity of gut microbiome features. However, we also reported per species analysis using a generalized model as this is more commonly used method. Please note that the findings between these two methods can differ substantially since the constructed model with one included bacterial species can be substantially different from a model with all included bacterial species. Lastly, it is possible that the gut microbiome associations we observed are merely covariation with HRQoL and general health (which is known to be associated with the gut microbiome[15,21]). However, the gut microbiome could be a potentially modifiable factor to increase HRQoL although, it should be noted that the effect sizes are moderate or the gut microbiome could function as a potential biomarker for KTR at risk for a low HRQoL. This cannot be disentangled with the current study design. More controlled studies are needed to further examine the relationship between the gut microbiome in HRQoL in KTR. Future studies that aim to study HRQoL and the gut microbiome in an experimental setting should use methods such as real time PCR to better approximate bacterial count to properly characterize the effect size as we were constrained by relative abundance from metagenomic sequencing data in the current study[38]. In such a study, supplementation of butyrate producing species in the form of probiotics could considered as we found a lower abundance of butyrate producing species and pathways in KTR with lower HRQoL.

In conclusion, gut microbial features are significantly associated with HRQoL features in KTR. Both summary scores of the physical and mental domains of HRQoL were significantly associated with composition of the gut microbiome. In addition, bacterial species that encompass principal component one were all, with strikingly consistent directionality, previously associated with unhealthy individuals (with disease[21]). While the observational study design does not allow us to analyze causality, we provided a comprehensive overview of the associations between the gut microbiome and HRQoL while adjusting for confounders. Our work will aid future studies in selecting potential modifiable gut microbial factors which could potentially improve the HRQoL for KTR.

## Methods

### Study design

All cross-sectional data of KTR from the ongoing, prospective, TransplantLines Biobank and Cohort study[14] (Trial registration number NCT03272841) that provided a fecal sample was included. A detailed description of the TransplantLines study has been published previously[14]. Briefly, from June 2015 all (potential) adult solid organ transplant recipients and kidney donors at the University Medical Center Groningen (UMCG), The Netherlands, were invited to participate. All feces samples collected by KTR till August 2019 were analyzed ($n = 751$). Of this cohort of KTR, HRQoL data were available in 507 KTR.

In order to calculate gut microbial distance to general population controls, 1183 age-, sex- and BMI-matched control subjects from the DMP were used[21]. Fecal samples from TransplantLines and DMP were processed with the same DNA extraction protocols and sequencing platform (see below). All participants signed an informed consent form prior to sample collection. TransplantLines (METc 2014/077) and Lifelines (METc 2017/152) were approved by the local institutional ethics review board (IRB) from the UMCG. Both studies adhere to the UMCG Biobank Regulation and are in accordance with the World Medical Association (WMA) Declaration of Helsinki and the Declaration of Istanbul.

### Phenotypic data

Demographic and clinical data were extracted from the patient files. Diabetes was defined according to criteria of the American Diabetes Association[39], and eGFR was calculated using the Chronic Kidney Disease Epidemiology Collaboration equation[40]. During a study visit, medication use was verified, anthropometrics were measured, clinical frailty was scored using the clinical frailty scale and hand grip strength was measured. Clinical frailty scores were classified in three groups: clinically well (score 1 to 3), clinically vulnerable (score 4) and clinically frail (score 5 or higher). According to the TransplantLines Biobank and Cohort' study design, additional physical assessments were performed in a part of the participants, among which the 2-min walk test, 4-meter walk test, timed-up-and-go test and sit-to-stand stand test. All assessments were described in detail previously[14]. Patient reported outcome measurements were assessed using questionnaires. HRQoL was assessed using the well-validated SF-36, which assesses HRQoL by 36 questions and results in domain scores of which a physical and mental component score, reflecting physical and mental HRQoL, can be calculated[17,18]. The PCS was calculated by taking the average of the general health, physical health, role limitations due to impairment of physical health and pain scores. The MCS was calculated by taking the average score of the emotional well-being, role limitations due to emotional problems, impaired social functioning and impaired vitality scores. Feeling of fatigue and anxiety were assessed using the CIS20R[41,42] and STAI6[43], respectively. Severe fatigue was defined as a score >= 35 on the subscale 'fatigue severity'[42], and anxiety was defined as a score > 50[44]. Depressive symptoms were assessed using the PHQ-9, and major depression was defined as a score >= 10[45].

### Sample selection and gut microbiome data generation

**Fecal sample collection and subsequent processing.** Patients were asked to collect a fecal sample the day prior to the study visit. A FecesCatcher (TAG Hemi VOF, Zeijen, The Netherlands) was sent to the patients at home. Feces were collected and stored in appropriate tubes and frozen at home (at −18 °C) immediately after collection. The participant transported the frozen fecal sample in cold storage (with ice cubes or in a cooler) to the study visit the following day. Subsequently, the fecal sample was immediately stored at −80 °C. Participants in the DMP project produced, collected, and froze fecal samples at home

using standardized stool collection kits provided by the UMCG. At home frozen (at −18 °C) fecal samples were collected by UMCG personnel and transported on dry ice and stored at UMCG at −80 °C until DNA extraction. Stool water content was analyzed by freeze-drying for 48 h under 0.5 bar at −50 °C.

**DNA extraction.** Microbial DNA was extracted using QIAamp Fast DNA Stool Mini Kit (Qiagen, Germany) according to the manufacturer's instructions. The QIAcube (Qiagen, Germany) automated sample preparation system was used for this purpose. Library preparation was performed using NEBNext® Ultra™ DNA Library Prep Kit for Illumina for samples with total DNA amount <200 ng, as measured using Qubit 4 Fluorometer, while samples with DNA yield >200 ng were prepared using NEBNext® Ultra™ II DNA Library Prep Kit for Illumina®. Libraries were prepared according to the manufacturer's instructions. Metagenomic shotgun sequencing was performed using Illumina HiSeq 2000 sequencing platform and generated -8 Gb of 150 bp paired-end reads per sample (mean 7.9 gb, st.dev 1.2 gb). Library preparation and sequencing were performed at Novogene, China.

**Metagenomic data processing.** Illumina adapters and low-quality reads (Phred score < 30) were filtered out using KneadData (trimmatic options: "LEADING:20 TRAILING:20 SLIDINGWINDOW:4:20 MINLEN:50"; v0.5.1)[46]. Then Bowtie2 (default settings, v2.3.4.1)[47] was used to remove reads aligned to the human genome (hg19). The quality of the reads was examined using FastQC toolkit (v0.11.7) with a minimal read depth of 10 million reads after quality control. Taxonomy alignment was done by MetaPhlAn2 (default settings, v2.7.2)[47,48] with the database of marker genes mpa_v20_m200. MetaCyc pathways were profiled by HUMAnN2 (default settings, v0.11.1)[49]. KEGG orthologs were obtained from MetaCyc using the humann_regroup_table script. Next, we used the *omixerRPM r-package (v0.3.2)* to reclassify KEGG orthologs into gut brain modules.[11] Samples were further excluded in case of a eukaryotic or viral abundance >25% of total microbiome content or a total read depth <10 million. In total, we identified 1132 taxa (17 phyla, 27 class, 52 order, 98 family, 231 genera and 705 species) and 586 metabolic pathways. After filtering for a prevalence of 10% and relative abundance threshold of 0.01%, 384 taxa (8 phyla, 14 class, 20 order, 40 family, 83 genera and 219 species) and 351 metabolic pathways. Hereafter, total-sum normalization was applied. Analyses were performed using locally installed tools and databases on CentOS (release 6.9) on the high-performance computing infrastructure available at UMCG and University of Groningen (RUG). An example of scripts used for microbiome process is available at https://github.com/GRONINGEN-MICROBIOME-CENTRE/TransplantLines.

**Statistical analysis**
Centered log-ratio normalization was used due to the compositional nature of the metagenomic sequencing data[19]. We used two times the minimum relative abundance as zero-imputation method for the centered log-ratio normalization. To assess differences between levels of categorical variables, we performed a Wilcoxon rank sum test in case of two levels and a kruskal wallis rank sum test in case of three or more levels. Associations between principal components and phenotypes were assessed using spearman correlation on all HRQoL measures. A Mann–Whitney *U*-test was used to assess differences between quartiles of HRQoL scores. PCA was performed using Euclidean distance between clr-transformed abundances (Aitchison distance[50]) of bacterial species. The Shannon diversity index was calculated using the *vegan (v2.6.5)*[51] package in R, of which also the ADONIS function was used with 9999 permutations to assess the proportion of explained variance for each phenotype on the Aitchison distance matrix.

Association analysis between HRQoL gut microbial features (species, metabolic pathways and gut brain modules) and was

performed using elastic net regression using the *caret (v6.0.91)* package in R and a generalized linear model (GLM) using the *stats (v4.1.3)* in R. Elastic net regression was performed to account for correlation between bacterial species and better accommodate the compositional ecosystem of the gut microbiome. We used a 10-fold cross-validation with 20 resampling iterations to train the prediction models for prediction of HRQoL and other phenotypes. In total, 75% of the dataset was used as a training set and 25% as a test set. Elastic net regression was then used with a generalized linear model (family = gaussian for numerical variables and family = binomial for categorical variables) and a tunelength of 30. All potential confounders (Age, sex, BMI, eGFR, stool water content, antibiotics, antihypertensive treatment, diabetes, dialysis, laxative, proton pump inhibitors) that were available in the current study were included in the elastic net and GLM analysis. In the elastic net analysis confounders were not forces into the model but instead could be penalized. This approach was taken to find the best predictive model per HRQoL feature. We accounted for multiple testing using Benjami-Hochberg correction and an FDR < 0.05 was considered as significant.

## Reporting summary
Further information on research design is available in the Nature Portfolio Reporting Summary linked to this article.

## Data availability
The raw microbiome metagenomic sequencing data are publicly available at the NIH's Sequence Read Archive (SRA) under accession number PRJNA1035431. Due to patient confidentiality, the clinical data associated with the metagenomic datasets are not publicly available but can be made available upon request. Access to this clinical dataset requires a minimal access procedure consisting of a request per email (datarequest.transplantlines@umcg.nl). A response will be provided within 2 weeks. This access procedure is to ensure that the clinical data are being requested for research/scientific purposes only and thus complies with the informed consent signed by TransplantLines participants, which specifies that the collected data will not be used by commercial parties.

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

## Acknowledgements

We would like to thank all participants from the TransplantLines- and Lifelines cohort and biobank study. We would like to thank the Center for Information Technology of the University of Groningen (RUG) for support and for providing access to the Peregrine high-performance computing cluster and the Genomic Coordination Center (UMCG and RUG) for support and for providing access to Calculon and Boxy high-performance

computing clusters. The TransplantLines Biobank and Cohort study received funding from Astellas BV (TransplantLines Biobank and Cohort study) and Chiesi Pharmaceuticals BV (PA-SP/PRJ–2020-9136) and was co-financed by the Dutch Ministry of Economic Affairs and Climate Policy by means of the PPP-allowance made available by the Top Sector Life Sciences & Health to stimulate public-private partnerships. Sequencing of the kidney part of the TransplantLines cohort was funded by a grant from the Dutch NWO/TTW/DSM partnership program Animal Nutrition and Health (project number 14939) to S.J.L.B. R.K.W. is supported by the Seerave Foundation, the Netherlands Organization for Scientific Research (NWO), and the EU Horizon Europe Program grant miGut-Health: personalized blueprint of intestinal health (101095470).

## Author contributions

R.K.W. and S.J.L.B. designed the study. J.C.S., T.J.K., R.M.D., M.H.d.B., V.E.dM., H.B., S.P.B. and S.J.L.B. gathered and prepared the data. T.J.K., J.R.B., and J.C.S. analyzed the data. J.R.B., T.J.K. and J.C.S. wrote the manuscript. R.G., A.V.V. and J.R.B. provided advice for statistical methods. R.G., L.M.N., S.Z., A.V.V., D.K., R.M.D, A.P., E.E.Q., R.A.P., B.H.J., M.H.d.B., V.E.d.M., H.B., S.P.B., E.A.M.F., A.Z., J.F., H.J.M.H., S.J.L.B. and R.K.W. were involved in critically reviewing the manuscript.

## Competing interests

The authors declare no competing interests.

## Additional information

[1]Department of Gastroenterology and Hepatology, University of Groningen, University Medical Center Groningen, Groningen, the Netherlands. [2]Department of Internal Medicine, Division of Nephrology, University of Groningen, University Medical Center Groningen, Groningen, the Netherlands. [3]Department of Genetics, University of Groningen, University Medical Center Groningen, Groningen, the Netherlands. [4]Department of Surgery, division of Transplantation Surgery, University of Groningen, University Medical Center Groningen, Groningen, the Netherlands. [5]Department of Surgery, section of Hepatobiliary Surgery and Liver Transplantation, University of Groningen, University Medical Center Groningen, Groningen, the Netherlands. [6]Department of Pediatrics, University of Groningen, University Medical Center Groningen, Groningen, the Netherlands. [7]Department of Medical Microbiology and Infection prevention, University of Groningen, University Medical Center Groningen, Groningen, the Netherlands. [20]These authors contributed equally: J. Casper Swarte, Tim J. Knobbe. [21]These authors jointly supervised this work: Stephan J. L. Bakker, Rinse K. Weersma. ✉e-mail: r.k.weersma@umcg.nl

## TransplantLines investigators

C. Annema[8], F. A. J. A. Bodewes[6], M. T. de Boer[5], K. Damman[9], A. Diepstra[10], G. Dijkstra[1], C. S. E. Doorenbos[2], M. F. Eisenga[2], M. E. Erasmus[4], C. T. Gan[11], A. W. Gomes Neto[2], E. Hak[12], B. G. Hepkema[13], F. Klont[12], H. G. D. Leuvenink[4], W. S. Lexmond[6], G. J. Nieuwenhuis-Moeke[14], H. G. M. Niesters[6], L. J. van Pelt[13], A. V. Ranchor[15], J. S. F. Sanders[2], M. J. Siebelink[16], R. J. H. J. A. Slart[17], D. J. Touw[18], M. C. van den Heuvel[10], C. van Leer-Buter[19], M. van Londen[2], E. A. M. Verschuuren[11] & M. J. Vos[13]

[8]Department of Health Sciences, Section of Nursing Science, University of Groningen, University Medical Center Groningen, Groningen, the Netherlands. [9]Department of Cardiology, University of Groningen, University Medical Center Groningen, Groningen, the Netherlands. [10]Department of Pathology & Medical Biology, University of Groningen, University Medical Center Groningen, Groningen, the Netherlands. [11]Department of Respiratory Diseases, Tuberculosis and Lung Transplantation, University of Groningen, University Medical Center Groningen, Groningen, the Netherlands. [12]Unit of PharmacoTherapy, -Epidemiology & -Economics, Groningen Research Institute of Pharmacy, University Medical Center Groningen, University of Groningen, University Medical Center Groningen, Groningen, the Netherlands. [13]Department of Laboratory Medicine, Transplantation Immunology, University of Groningen, University Medical Center Groningen, Groningen, the Netherlands. [14]Department of Anesthesiology, University of Groningen, University Medical Center Groningen, Groningen, the Netherlands. [15]Department of Health Psychology, University of Groningen, University Medical Center Groningen, Groningen, the Netherlands. [16]University Medical Center Groningen Transplant Center, University of Groningen, University Medical Center Groningen, Groningen, the Netherlands. [17]Medical Imaging Centre, Department of Nuclear Medicine and Molecular Imaging (EB50), University Medical Center Groningen, University of Groningen, University Medical Center Groningen, Groningen, the Netherlands. [18]Department of Clinical Pharmacy and Pharmacology, University of Groningen, University Medical Center Groningen, Groningen, the Netherlands. [19]Department of Virology, University of Groningen, University Medical Center Groningen, Groningen, the Netherlands.

