## [Peer Review File · Nature Communications]

REVIEWER EXPERTISE

Reviewer #1. Microbiome / neuroactive potential / metagenomics.

Reviewer #2. Kidney disease / transplantation / Clinical.

Reviewer #3. Microbiome / gut-brain axis.

REVIEWER COMMENTS

Reviewer #1 (Remarks to the Author):

Swarte et al analysed microbiome composition (shotgun metagenomic data) in 507 fecal samples from kidney transplant recipients and its link with HRQoL.

Major comments

- The authors assess associations between gut microbiome composition and HRQoL quartiles in KTR, and introduce a general population afterwards. It would be useful to contextualize the analyses by comparing HRQoL of the KTR population to that of the general one, and then see if the associations described also hold in the larger set. In this way, it would also be possible to disentangle HRQoL associations from KTR-specific signals.

- The authors state they account for previously established microbiome covariates and show in Supplementary Table 2 the association between some covariates and RAND scores. However, both the list of potential confounders (why was gastrointestinal transit time or stool consistency – major covariate in many datasets - not included?) and the determination of potential confounding are incomplete: the authors don't test the association between covariates and microbiome composition to identify potential confounding, and the results of the associations with RAND scores aren't used to determine which covariates are included in the analysis. Also, it is not clear in what way covariates are introduced in the elastic net regularization approach. To make sure the results are solid, please provide an accurate identification of potential confounders and include them in the statistical models accordingly.

- What was the rationale to discretize HRQoL scores into quantiles? This is not included in the validated scoring instructions. Do the results hold if the data isn't grouped this way? Similarly,

although elastic net regularization has some advantages in specific cases, do the results hold if a simple correlation accounting for covariates is used?

- The methods section lacks important detail. This together with the sequencing data and metadata not being available unless a request is made to EGA makes it impossible to replicate the findings. In addition, the software the authors used for taxonomic and functional profiling are rather outdated: the MetaPhlAn2 was published in 2015 while MetaPhlAn4 was recently published, and HUMAnN2 in 2018 while HUMAnN3 was published in 2021.

Minor comments

- The introduction should be revised for accuracy. For example, in “The gut microbiome was recently identified as a determinant of HRQoL [...]” the study cited was cross-sectional, so causality could not be inferred. Or in “Multiple microbial species [...] appear to be associated [...]” the study performed analyses at the genus level, so species associations were not described.

- The supplementary tables need to be revised. Supplementary Table 1 has as a title “Characteristics of RTR”. Shouldn’t this be “KTR”? Also, Supplementary Table 6 has only been partially translated from Dutch to English (e.g. Sex=Vrouw, or using “yes” or “ja” in different rows).

- It looks like the MCS and PCS score were calculated in a custom way (i.e. averaging single scores), instead of using the instructions of the validated questionnaires including population norms. What was the reason for this?

- The finding that participants less physically active or who reported severe fatigue are more distant to the general population in terms of beta diversity is unexpected. The authors should comment on this in the discussion.

- Missing details in the methodology include: What was the minimum sequencing depth after quality control to include a sample in the analysis? What parameters were used in the different software? Computation of gut-brain modules from MetaCyc pathways is completely omitted. What method was used for imputation of zeroes in clr normalization? Versions of R packages are missing.

Reviewer #2 (Remarks to the Author):

Thank you for the opportunity to review this interesting work. This manuscript by Swarte and colleagues addresses the very relevant topic of the gut microbiome and health-related quality of life in kidney transplant recipients. The authors should be commended on this impressive work. However, I have some concerns:

Major concerns

- The authors state that the microbiome can be seen as a potential therapeutic target to improve QoL. This precludes a causal relationship between gut dysbiosis and impaired QoL. Is this associative study enough to conclude on causality? Luckily, the authors come back to this in the discussion – the conclusion in the abstract is too firm.
- What about the influence of diet in the observed association? In the original paper on the design of the Transplantlines, diet and lifestyle are included as variables. I am wondering why diet was not taken into account in this particular manuscript. The paper would certainly increase its clinical relevance when diet is taken into account; I strongly suggest to include this, and at least to pay attention to it in the discussion. Now, diet and its importance, remains a big question mark throughout the paper.

Minor concerns

- Figure 3 is not clear to me. The legend says that it should be a heatmap, but there is only color for antihypertensive treatment and the 4 meter walking test?
- Why the authors focused on a relatively late period after transplantation? Also, the range of time after transplantation is quite broad. This merits some words in the introduction.
- A detailed description of the use of antibiotic therapy is lacking.

Reviewer #3 (Remarks to the Author):

In this manuscript, Swarte and colleagues examined the gut microbiome and quality of life in a sample of 507 kidney transplant recipients (KTR), finding some small but interesting associations between these measures. There is certainly research interest in the relationship between gut microbiome factors and physical and mental health outcomes, so the topic is of importance, and the writing is clear. I have some queries for the authors, detailed below.

One major concern is that there seemed to be a substantially different collection protocol for the KTR participants and the control participants from the DMP project. In the methods, it states that KTR participants collected and froze samples at home before transport to the facility the following day. The average and range of transportation times from home to study visit should be reported if available. In contrast, DMP samples were collected from participant homes within 15 minutes of production and transported on dry ice. The former method less much more room for variation and potentially an additional freeze-thaw cycle, which is known to impact microbiome composition. The differences in methodology and the implications for data interpretation must be clearly stated in the results and discussion.

For similar reasons, it is important to clarify whether the DNA extraction and metagenomic sequencing were conducted at the same time for the different sets of samples or if samples were analyzed in separate batches (both for the extraction and sequencing steps). If separate, were any samples re-sequenced to adjust for run effects?

In the statistical analysis, a number of covariates were used, reflecting the importance of controlling for potential confounds where possible. However, it is also common to see such analyses presented both with and without covariates, for comparison against other studies where covariates are not included or where different covariates are used, and for transparency so that the reader can understand the “raw” or direct relationships between variables of interest. I would suggest that this could be included as a supplement.

Some of the approaches to the microbiome analysis were rather unusual and perhaps need further explanation. For example, why did the authors decide to look at correlations with PC1? Was this based on previously conducted studies or a theoretical rationale? This is a potential concern given that, in their sample, PC1 explained only 7% of the variance in the microbiome (which seems substantially smaller than what I have typically seen in the literature). Can the authors offer any potential reasons for the low amount of variance explained and provide some discussion on how meaningful correlations are between this subset of microbiome data and the other measures?

I am also unfamiliar with elastic-net regularization and it is probably worth explaining the advantage of this approach over more commonly used approaches (e.g. HUMAnN or MaAsLin).

It was not clear to me why quartiles were used to analyze the HRQoL data, rather than correlations. Can the authors provide some justification for this?

Typically, several measures of alpha diversity are analyzed, reflecting the different aspects of diversity that they measure. Since the Shannon index incorporates evenness, it is probably worthwhile to examine a measure of richness as well.

For the beta diversity plots in Figures 1A & B, it seems unusual to represent the groups by their centroid (at least, I am assuming that's what the bold dots represent) rather than ellipses to illustrate the overlap between groups. I am also a little confused about the analysis of beta diversity. Was PERMANOVA carried out to determine group differences between the quartiles based on the overall data, or were data analyzed only for each principal component separately? Again, this seems quite unusual and further explanation would be informative.

In general, there is no mention of how the authors accounted for multiple comparisons in the methods of the analyses (including how FDR was calculated).

Given that the GBMs were developed to test for potential gut-brain links, it seems surprising that there were many more GBMs related to physical HRQoL than mental HRQoL (in fact, if I interpreted the data correctly, there was only 1 GBM that uniquely associated with mental health?). This is perhaps worth considering in the discussion.

In the discussion the authors focus on butyrate, yet there was no mention of butyrate pathways in the GBM analysis. Were there any significant findings for butyrate in the predictive functional analysis? If not, can the authors reconcile this with their hypotheses on butyrate-producing bacteria?

Overall, the effect sizes seem relatively small, which should be acknowledged when considering whether the microbiome will be a valuable treatment target as proposed.

Minor points:

- For the HRQoL measures, it would be good to know (expected) averages for the general population or standardized bands of functioning if they exist, to allow readers to interpret the averages for this specific population of KTR.
- It should be noted as a limitation that mental health was assessed only through self-report.
- Be careful of language that may be interpreted as causal or directional (e.g. line 121: HRQoL “explained” interindividual variation in the gut microbiome)
- Line 342: typo “known to covariate with”

- The recommendation that probiotics could potentially benefit HRQoL seems a bit general – what species might be good candidates to test in this regard, or are there metabolites that could be used as supplements instead?
- It would be helpful if page/line references were included in the STORMS checklist
- It would be helpful if acronyms could be expanded in the supplementary tables (or at least include a key in the index page)

Reviewer comments

Reviewer #1 (Remarks to the Author):

Swarte et al analyzed microbiome composition (shotgun metagenomic data) in 507 fecal samples from kidney transplant recipients and its link with HRQoL.

Response:

We thank the reviewer for the time and effort invested in reviewing our manuscript.

Major comments

Comment 1.0: The authors assess associations between gut microbiome composition and HRQoL quartiles in KTR, and introduce a general population afterwards. It would be useful to contextualize the analyses by comparing HRQoL of the KTR population to that of the general one, and then see if the associations described also hold in the larger set. In this way, it would also be possible to disentangle HRQoL associations from KTR-specific signals.

Response 1.0: We agree with the reviewer that it would be insightful to disentangle any KTR-specific HRQoL microbiome signal. We now included HRQoL and microbiome data of 151 healthy controls from the TransplantLines cohort and biobank study.¹ These control samples are unrelated healthy individuals who were screened to be potential kidney donors (unrelated to the 507 KTR, but from the same geographic area) that were included prior to kidney donation. They are in good health as they were deemed fit for kidney donation. These samples were included in the current study to give the reader an understanding of how HRQoL is different between KTR and healthy controls and help disentangle KTR specific signals. All HRQoL features were significantly different between KTR and healthy controls (Wilcoxon, $P < 1.43 \times 10^{-7}$; **Figure 1, Supplementary Figure 2**). We compared HRQoL and the gut microbiome for these 151 healthy controls. This analysis identified KTR specific signals. For gut microbial species, one association was identified between four of the physical component scale (PCS), six associations for physical functioning and six associations for general health attribution were also observed in the healthy controls (**Supplementary Table 6**). For metabolic pathways, we observed five pathways that were associated with the PCS that were also observed in the healthy controls (**Supplementary Table 8**). Finally, for GBM, two associations with vitality and seven associations with general health attribution were also observed in the healthy controls (**Supplementary Table 10**). These new analyses were added in **lines 100-102, 111-119, 223-232, 241-246 and 286-290, Supplementary Table 6, 8, and 10 and Figure 3, main Figure 3**. Furthermore, we have now also added an extensive comparison with the work of Valles-Colomer et al. in **lines 345-357** of the discussion.²

We included 1,183 gut microbiome profiles from the general population – the so-called Dutch Microbiome Project (DMP) – in the current study for the purpose of quantifying microbiome dysbiosis in the KTR population.³ However, because HRQoL data was not collected, it is not available for these participants.⁴

Figure 1, Supplementary figure 2 HRQoL for KTR and healthy controls. Each boxplot and violin plot represents HRQoL features from the SF-36-questionnaire for KTR in blue and 151 healthy controls from the TransplantLines cohort in gray. All HRQoL features were significantly different between KTR and healthy controls (Wilcoxon, $P < 1.43 \times 10^{-7}$)

Comment 1.1: The authors state they account for previously established microbiome covariates and show in Supplementary Table 2 the association between some covariates and RAND scores. However, both the list of potential confounders (why was gastrointestinal transit time or stool consistency – major covariate in many datasets - not included?) and the determination of potential confounding are incomplete: the authors don't test the association between covariates and microbiome composition to identify potential confounding, and the results of the associations with RAND scores aren't used to determine which covariates are included in the analysis. Also, it is not clear in what way covariates are introduced in the elastic net regularization approach. To make sure the results are solid, please provide an accurate identification of potential confounders and include them in the statistical models accordingly.

Response 1.1: We have added a list of potential confounders in **lines 229-230** of the results section and **lines 526-527** of the methods section. The Bristol stool chart which is commonly used to assess transit time was unfortunately not available for all patients in the current study. However, we measured stool water content for all included samples, which provide an even more objective marker for stool consistency and transit time than the Bristol Stool Scale.⁵⁻⁷ In the revised manuscript, we have included stool water content in all of our analyses. In the permutational multivariate ANOVA (PERMANOVA) analysis, stool water content was not significantly associated with any of the SF36-scores ($P > 0.05$) but did significantly explain variation in the gut microbiome ($R^2 = 0.7\%$, $P = 1.0 \times 10^{-4}$) and it was significantly correlated with the Shannon diversity index ($r = -0.12$, $P = 5.0 \times 10^{-3}$). We have added these results to the manuscript (**lines 93-102** and **Supplementary Table 2 and 4**). Other potential confounders considered in the current manuscript are: age, sex, BMI, eGFR, antibiotics, antihypertensive treatment, diabetes, dialysis, laxative, proton pump inhibitors of which a complete list was now added in **lines 226-232** of the results section and **lines 470-471** of the methods section. In the elastic net regularization model a large number of potentially correlated predictor variables were included: 219 microbial species, 351 pathways and 56 gut brain modules and age, sex, BMI, eGFR, stool water content, antibiotics, antihypertensive treatment, diabetes, dialysis, laxative, proton pump inhibitors. This approach uses regularization to select the predictor variables that best explain the data while accounting intercorrelation between predictors.⁸ Please note that we have updated all results and now also included stool water content.

Comment 1.2: What was the rationale to discretize HRQoL scores into quantiles? This is not included in the validated scoring instructions. Do the results hold if the data isn't grouped this way? Similarly, although elastic net regularization has some advantages in specific cases, do the results hold if a simple correlation accounting for covariates is used?

Response 1.2: Indeed, dividing HRQoL scores into quartiles is not included in the validated scoring instructions. However, quartiles were only used for visualization purposes (**Main Figure 1A and 1B**). In all association analyses, the raw SF36 scores (i.e. not divided into quartiles) were analyzed. We have now also added the results from a correlation analysis between the raw SF36 scores and the principal components to the results section (**lines 160-161; Supplementary Table 3**).

The main advantage of Elastic Net over simple linear regression is that it incorporates both L1 and L2 regularization and can therefore model a large number of potentially correlated predictor variables while minimizing overfitting and reducing bias due to multicollinearity.⁸

We fully understand and agree that readers also might be interested into per species analysis with and without confounders. Therefore, we have now added the results of an analysis where we model each microbial species individually with and without confounders. More specifically, we including three models (1) no confounders; (2) microbiome related confounders, i.e. stool water content, antibiotics, laxatives, proton pump inhibitors, and (3) all identified confounders, i.e. Age, sex, BMI, eGFR, stool water content, antibiotics, antihypertensive treatment, diabetes, dialysis, laxative, proton pump inhibitors. Please note, however, that the results from this analysis compared with the Elastic Net can differ substantially due overfitting and multicollinearity. Additionally, multiple hypothesis correction heavily reduces the number significant association; modeling individual species with and without confounders represents a large number of hypotheses. More specifically, with 20 phenotypes and 10 confounders ($10 + 20 = 30$), 217 microbial species, 350 metabolic pathways and 56 gut brain modules, we need to correct for 16,540 hypotheses tested ($[30 \times 217] + [30 \times 350] + [30 \times 56] = 16,540$). Thus, many results do not pass a false discovery rate (FDR) threshold of 0.05 or 0.10. For this reason, we have decided to keep and focus on the results from the Elastic Net model in the main text but have still added the results from the per species analysis to the **lines 223-232, 241-246 and 286-290** of the results section and **Supplementary Tables 7, 9 and 11**. We also added a paragraph where we discuss the strengths and limitations of the Elastic Net and simple linear regression to the discussion (**lines 390-397**).

Comment 1.3: The methods section lacks important detail. This together with the sequencing data and metadata not being available unless a request is made to EGA makes it impossible to replicate the findings. In addition, the software the authors used for taxonomic and functional profiling are rather outdated: the MetaPhlAn2 was published in 2015 while MetaPhlAn4 was recently published, and HUMAnN2 in 2018 while HUMAnN3 was published in 2021.

Response 1.3: We have added more details to the methods section by including additional information regarding the metagenomic data processing and statistical analyses (**lines 484-493 and lines 504-530**). We also included missing details reported by the reviewer in comment 1.8.

We would like to stress that all data is available upon request through the European Genome-phenome Archive (EGA). We cannot share our data publicly given the depth and detail of clinical data used in this study. In adherence to the participants' privacy as stated by The General Data Protection Regulation 2016/679 and TransplantLines Biobank and Cohort Study regulations, we are not allowed to make phenotype data publicly available. Additionally, we have provided all our codes and mock data to GitHub (<https://github.com/GRONINGEN->

MICROBIOME-CENTRE/TransplantLines) so that the reader can better follow how the analysis was performed.

At the time of data processing, we used MetaPhlan2 and HUMAnN2 because MetaPhlan3 and HUMAnN3 were not yet available. We would like to refrain from reprocessing all data to MetaPhlan4 and HUMAnN3 because we believe the benefits does not supersede the costs. Our main reasons are: (1) The TransplantLines cohort study and biobank follows the same processing protocols as the Dutch Microbiome Project (Gacesa et al. 2022). This enables us to compare our transplantation cohort to a larger sample of the (same) general population. In fact, some of our main findings come from comparing our KTR results with the findings of the DMP ⁴; (2) due to the substantial updates in the MetaPhlan database, comparing MetaPhlan4 results to the DMP, including other studies that used MetaPhlan2 or 3 is not possible (a large fraction of the species labels have changed in MetaPhlan4); (3) updating and re-analyzing all of our data would take a substantial amount of time. That said, our team in the Groningen Microbiome Hub is currently working on updating all of our team's microbiome data (~21,000 metagenomes) to the most recent version of bioBakery. This is indeed a big team effort that takes considerable time and comes with large computational costs. Given the substantial amount of time it would take to use MetaPhlan4 and HUMAnN3 in the current study and reanalyze all data compared to the expected limited increase of potential significant findings and decreased comparability with the DMP we would like to refrain from reanalyzing all data in the current study.

Minor comments

Comment 1.4: The introduction should be revised for accuracy. For example, in “The gut microbiome was recently identified as a determinant of HRQoL [...]” the study cited was cross-sectional, so causality could not be inferred. Or in “Multiple microbial species [...] appear to be associated [...]” the study performed analyses at the genus level, so species associations were not described.

Response 1.4: We thank the reviewer for pointing this out. To accommodate the comment of the reviewer, we changed “The gut microbiome was recently identified as a determinant of HRQoL [...]” into “Recently, a cross-sectional study in the general population identified the gut microbiome as a factor associated with HRQoL.” In addition, we changed “Multiple microbial species [...] appear to be associated [...]” into “Multiple genera [...] appear to be associated [...]”.

Comment 1.5: The supplementary tables need to be revised. Supplementary Table 1 has as a title “Characteristics of RTR”. Shouldn’t this be “KTR”? Also, Supplementary Table 6 has only been partially translated from Dutch to English (e.g. Sex=Vrouw, or using “yes” or “ja” in different rows).

Response 1.5: We revised the supplementary tables. We changed the title of **Supplementary Table 1** into “Characteristics of KTR”, and translated in **Supplementary Table 6** the Dutch words to English.

Comment 1.6: It looks like the MCS and PCS score were calculated in a custom way (i.e. averaging single scores), instead of using the instructions of the validated questionnaires including population norms. What was the reason for this?

Response 1.6: Multiple calculations are commonly used for SF-36 data.¹³ We calculated the MCS and PCS scores using the average of the subscale scores, which is a method based on the scoring instructions and which has been used previously for the purpose of identification of determinants of HRQoL.^{14–17} The most important reason for using this method, and to not use the standardized method, is that there is no good reference population to standardize the MCS and PCS score. Ideally, this should be a population of kidney transplantation recipients from the Netherlands. Instead, our applied method is not influenced by a reference population. Furthermore, the purpose of our study was to understand the link between the gut microbiome and HRQoL, rather than comparing HRQoL with other populations. Therefore, the applied method appears most suitable for our study purpose. In addition, the scores resulting from the applied method are easier to understand where the scores are derived from compared to the standardized method, as it is a very straightforward approach - four physical components are used to calculate the physical component score. For the standardized PCS score, all subscales are considered, and each subdomain gets a negative or positive weighing. This complicates the interpretation of the results.

To accommodate the comment of the reviewer, we added sensitivity analyses using the standardized PCS and MCS score, based on the United States population means, standard deviations and factor score coefficients as published in the scoring instructions in this point-by-point response (**Table 1, Figure 2**).¹⁸ Statistics based on standardized scores is included in **lines 113-119** of the results section. The principal component analyses and the correlations with the distance to the general population control using the standardized score were similar compared with the unstandardized scores and thus not included in the manuscript.

Figure 2 Disease-associated bacterial species in the gut microbiome of KTR are associated with standardized HRQoL scores. **(A & B)** Principal component analysis on the clr-transformed species reflecting the Aitchison distances between KTR. The standardized physical component score (A) and the standardized mental component score (B) are divided into quartiles (KTR were divided into quartiles (Q1, Q2, Q3, Q4) based on their standardized PCS and standardized MCS with Q1 containing the lowest HRQoL scores and Q4 the highest HRQoL scores). The large dots represent centroids per group and the dashed circles represent 95% confidence ellipses. **(C & D)** Correlation plot between the standardized PCS (C), the standardized MCS (D) and the distance to general population controls. This dysbiosis score was calculated previously by calculating the Aitchison distance from KTR with 1183 age-, sex- and BMI-matched general population controls³.

Table 1 Spearman correlations between the standardized PCS, MCS and PC's

	Standardized PCS		Standardized MCS	
	r	P-value	r	P-value
PC1	0.18	6.13x10 ⁻⁵	0.08	0.09
PC2	-0.02	0.60	-0.01	0.92
PC3	0.09	0.04	0.03	0.57
PC4	-0.06	0.02	-0.01	0.88
PC5	0.18	6.13x10 ⁻⁵	0.01	0.91

Comment 1.7: The finding that participants less physically active or who reported severe fatigue are more distant to the general population in terms of beta diversity is unexpected. The authors should comment on this in the discussion.

Response 1.7: We hypothesize that these patients might have worse overall health, which is reflected in lower physical activity, higher frailty score and more severe fatigue. Therefore, these participants might be more exposed to factors that can worsen dysbiosis such as polypharmacy, antibiotic use and comorbidities. We have added this to our discussion in **lines 370-378**.

Comment 1.8: Missing details in the methodology include: What was the minimum sequencing depth after quality control to include a sample in the analysis? What parameters were used in the different software? Computation of gut-brain modules from MetaCyc pathways is completely omitted. What method was used for imputation of zeroes in clr normalization? Versions of R packages are missing.

Response 1.8: We have added all the missing details regarding the methodology in the methods section and would like to thank the reviewer for highlighting these missing details. Minimal read depth after quality control was 10 million reads, as now mentioned in **line 487**. KneadData used human genome for decontamination, trimmomatic options "LEADING:20 TRAILING:20 SLIDINGWINDOW:4:20 MINLEN:50". Bowtie2, FastQC, MetaPhlan2 and HUMAnN2 were used with default settings. We obtained KEGG orthologs¹⁹ from MetaCyc pathways using the humann_regroup_table script from the HUMAnN2 software package. Next, we used the omixerRPM r-package (<https://github.com/omixer/omixer-rpmR>) to

reclassify KEGG orthologs into gut brain modules. We thank the reviewer for pointing out that we did not include this in the methods section and have now added this in **lines 491-493**.² We used two times the minimal relative abundance value for zero imputation in the CLR-transformation. Explanation was added in **lines 505-506** of the methods section. R-package versions are now mentioned in text.

Reviewer #2 (Remarks to the Author):

Comment 2.0: Thank you for the opportunity to review this interesting work. This manuscript by Swarte and colleagues addresses the very relevant topic of the gut microbiome and health-related quality of life in kidney transplant recipients. The authors should be commended on this impressive work. However, I have some concerns:

Response 2.0: We thank the reviewer for the kind words and comments, please find our detailed responses and revision below.

Major concerns

Comment 2.1: The authors state that the microbiome can be seen as a potential therapeutic target to improve QoL. This precludes a causal relationship between gut dysbiosis and impaired QoL. Is this associative study enough to conclude on causality? Luckily, the authors come back to this in the discussion – the conclusion in the abstract is too firm.

Response 2.1: We can indeed not make any causative claims. We have therefore toned down the concluding remarks in the abstract (**lines 42-45**).

Comment 2.1: What about the influence of diet in the observed association? In the original paper on the design of the TransplantLines, diet and lifestyle are included as variables. I am wondering why diet was not taken into account in this particular manuscript. The paper would certainly increase its clinical relevance when diet is taken into account; I strongly suggest to include this, and at least to pay attention to it in the discussion. Now, diet and its importance, remains a big question mark throughout the paper.

Response 2.1: In the TransplantLines cohort and biobank study, diet data is collected with food frequency questionnaires (FFQ).¹ However, the FFQ data has not yet been fully curated and processed in the TransplantLines project. With the curated data currently available from 430 (75%) of our KTR population, we were able to analyze the relationship between the gut microbiome and diet (macro nutrients) from FFQs. This data includes total caloric intake and the intake of fiber, mono- and disaccharides, polysaccharides and protein, cholesterol, unsaturated-, saturated and multi saturated fatty acids.

To analyze the relationship between diet - gut microbiome - HRQoL, we performed a mediation analysis. More specifically, we constructed a mediation model per macronutrient and microbial species using non-parametric bootstrapping with 1000 simulations. This was done with the *mediation* function from the *mediation* package in R. In this analysis,

macronutrients were considered as independent variables, gut microbial species were considered as mediators, and HRQoL features were considered as the dependent variables. However, after multiple testing correction, no significant mediation effects between macronutrients and HRQoL were identified (**FDR < 0.10, Table 2**). To account for multicollinearity, prevent overfitting and reduce the number of tests, we also performed a regularized mediation analysis. This analysis fits one mediation model using all macronutrients (as independent variables), all gut microbial species (as mediators) and all HRQoL features (as dependent variables) and then using regularization (similarly to the Elastic Net) to select the most relevant variables that explains the data. This was done using the *mvregmed.grid* function in the R package *regmed*. However, analysis did not identify any significant mediation effects either.

Given the negative findings and the amount of analyses and findings already presented in the manuscript, we now chose to not incorporate these findings into the current manuscript but instead devoted a section of our discussion in **lines 387-390** to address the potential role of diet in the observed associations between the gut microbiome and HRQoL. We would leave it to the discretion of the reviewer and editors whether to incorporate these mediation analyses in the main manuscript.

**Table 2 Results from mediation analysis between diet - HRQoL - gut microbiome.**

| | Total caloric intake | Fiber content | Mono and disaccharides | Poly-saccharide | Protein | Cholesterol | Saturated fatty acids | Unsaturated fatty acids | Multisaturated fatty acids |
|---|---|---|---|---|---|---|---|---|---|
| | Significant FDR: <0.10 (n) | | | | | | | | |
| **Physical component scale** | 0 | 0 | 0 | 0 | 0 | 0 | 0 | 0 | 0 |
| **Mental component scale** | 0 | 0 | 0 | 0 | 0 | 0 | 0 | 0 | 0 |
| **Physical functioning** | 0 | 0 | 0 | 0 | 0 | 0 | 0 | 0 | 0 |
| **Social functioning** | 0 | 0 | 0 | 0 | 0 | 0 | 0 | 0 | 0 |
| **Physical restrain** | 0 | 0 | 0 | 0 | 0 | 0 | 0 | 0 | 0 |
| **Emotional restrain** | 0 | 0 | 0 | 0 | 0 | 0 | 0 | 0 | 0 |
| **Mental health** | 0 | 0 | 0 | 0 | 0 | 0 | 0 | 0 | 0 |
| **Vitality** | 0 | 0 | 0 | 0 | 0 | 0 | 0 | 0 | 0 |
| **Pain** | 0 | 0 | 0 | 0 | 0 | 0 | 0 | 0 | 0 |
| **General health attribution** | 0 | 0 | 0 | 0 | 0 | 0 | 0 | 0 | 0 |

Minor concerns

Comment 2.2: Figure 3 is not clear to me. The legend says that it should be a heatmap, but there is only color for antihypertensive treatment and the 4-meter walking test?

Response 2.2: We are not sure if the reviewer had access to the complete Figure 3 (Figure 3 of rebuttal letter). The heatmap is made up of 4 different columns with results from the elastic-net analysis. The root-mean-square deviation (RMSE) column depicts the RMSE from the model for numerical variables. The accuracy column depicts the accuracy for the categorical variables. The phenotypes column consists of the betas of potential confounders and the species column depicts the betas of species. There are many more squares than antihypertensive treatment and the 4-meter walking test that have color while there are also gray squares that depict a beta of 0 in the elastic-net model. We hope this clarifies Figure 3 for the reviewer.

Figure 3, Main figure 3 of manuscript. Multiple bacterial species in the post-transplant gut microbiome are associated with HRQoL and assessments reflecting physical and mental health. This heatmap depicts the results from the elastic-net analysis for associations with a $\beta > 0.3$ or $\beta < 0.3$ that was performed for HRQoL scores and assessments of physical and

mental health. For numerical variables, the RMSE of each elastic-net model is shown in the first two columns (test and train set, respectively) and for categorical variables, the accuracy is depicted in the next two columns (test and train set, respectively). Selected features in the elastic-net model are colored by effect size. Gray tiles encompass phenotypes or species that were not included in the model. Plus-sign depict associations that were also observed in the control cohort. RMSE: Root-mean-square deviation, Accur.: accuracy.

Comment 2.3: Why the authors focused on a relatively late period after transplantation? Also, the range of time after transplantation is quite broad. This merits some words in the introduction.

Response 2.3: In the current study we used all cross-sectional data regarding HRQoL and the gut microbiome that was available in the TransplantLines cohort and biobank study. The time since transplantation (median time after transplantation was 5.0 years [IQR 1.0-12.0]) indeed had quite a broad range (**Figure 4**). However, we think that the current cohort is a good representation of KTR who were approximately more than one year after transplantation at the moment of inclusion. In our previous analysis regarding the gut microbiome of KTR we observed a large fluctuation of the gut microbiome within the first year after transplantation.³ We also chose for the cross-sectional cohort because the survival time of KTR is getting longer but HRQoL remains poor for some KTR.^{23,24} We have added the rationale for this decision in the introduction in **lines 52-53 and lines 74-76** and added **Figure 2** as **supplementary figure 1** to better show the distribution of time since transplantation in the current study.

Figure 4, Supplementary figure 1 of the main manuscript. Years since transplantation for KTR. Bar graph depicting years since transplantation count. Please note that there were 58 participants included between 10-12 months after transplantation. These 58 participants fall into the 0 years since transplantation bin in this graph.

Comment 2.4: A detailed description of the use of antibiotic therapy is lacking.

Response 2.4: We have added a detailed description of the type of antibiotic therapy to **Supplementary Table 1** for the 22 KTR that used antibiotics during sampling. In the analysis we chose to include a dichotomous variable regarding antibiotic use instead of a dichotomous variable per type of antibiotic, due to the low sample size numbers in antibiotic type variables (**Table 3**).

Table 3. Antibiotic use

Category	Characteristics	Categorical (n)	NA	Non missing
Medication	Antibiotics	22	0	507
Medication	Fluoroquinolone	5	0	507
Medication	Penicillin	2	0	507
Medication	Tetracycline	2	0	507
Medication	Aminoglycosides	1	0	507
Medication	Macrolides	4	0	507
Medication	Sulfonamides and trimethoprim	11	0	507

Reviewer #3 (Remarks to the Author):

Comment 3.0: In this manuscript, Swarte and colleagues examined the gut microbiome and quality of life in a sample of 507 kidney transplant recipients (KTR), finding some small but interesting associations between these measures. There is certainly research interest in the relationship between gut microbiome factors and physical and mental health outcomes, so the topic is of importance, and the writing is clear. I have some queries for the authors, detailed below.

Response 3.0: We thank the reviewer for the extensive comments and consideration of our current manuscript. A detailed response describing how we incorporated the suggestions of the reviewer can be found below.

Comment 3.1: One major concern is that there seemed to be a substantially different collection protocol for the KTR participants and the control participants from the DMP project. In the methods, it states that KTR participants collected and froze samples at home before transport to the facility the following day. The average and range of transportation times from home to study visit should be reported if available. In contrast, DMP samples were collected from participant homes within 15 minutes of production and transported on dry ice. The former method less much more room for variation and potentially an additional freeze-thaw cycle, which is known to impact microbiome composition. The differences in methodology and the implications for data interpretation must be clearly stated in the results and discussion.

Response 3.1: We would like to point out that the stool collection protocol in the TransplantLines project is very similar as the stool collection protocol in the Dutch microbiome project (DMP). Participants of both studies are asked to freeze the stool sample immediately after collection in their home freezer (-18°C). In both studies, stool samples are then transported within 24 hours to the hospital where they are stored at -80°C. The only difference between both studies is the method of transportation: in the TransplantLines study, participants use a special storage kit with cooling elements that were frozen at home (-18°C). These kits ensure that fecal samples remain completely frozen during transport. If a sample had thawed upon arrival at the hospital, the sample was not included in the biobank and excluded from the study. However, in both studies fecal samples were stored at home and collected and stored within 24 hours at -80°C in the hospital. In the DMP samples were not collected on dry ice within 15 minutes after defecation. We apologize for writing an ambiguous sentence in the methods section which we now have revised (**lines 467-471**).

To address the concern regarding the difference in transportation method and the potential influence in the observed associations in the current study, we would like to point out that we, in the current study, only used data from the DMP to calculate the distance to general population controls. However, to make sure that results we observe are not due to any difference that could be explained by the different freezing methods during transportation, we performed a sensitivity analysis. In this sensitivity analysis, we used 151 samples from healthy kidney donors (which are part of the TransplantLines study; see response #1.0) as controls. These samples were part of the same batch as the 507 KTR analyzed in the current study. All associations we report regarding the association between the distance to general population control remains when we use these 151 samples as the healthy controls except for general health attribution ($P < 0.05$, **Figure 4**). Because the 151 fecal samples from TransplantLines controls were collected in an exactly the same manner as the KTR, we are confident that the observed associations are caused by the very minor different freezing methods during transportation.

Figure 4 Correlations between dissimilarity metric of both general population controls from the Dutch microbiome project and controls from the TransplantLines project. Heatmap depicting significant (white boxes indicate a P-value > 0.05) spearman correlations of the dissimilarity metric calculated for both general population controls from the Dutch microbiome project and controls from the TransplantLines project. Red colors indicate positive correlations and blue indicate negative correlations

Comment 3.2: For similar reasons, it is important to clarify whether the DNA extraction and metagenomic sequencing were conducted at the same time for the different sets of samples or if samples were analyzed in separate batches (both for the extraction and sequencing steps). If separate, were any samples re-sequenced to adjust for run effects?

Response 3.2: DNA extract and metagenomic sequencing were done completely identical for the samples of both the TransplantLines and the Dutch microbiome project. All DNA extractions were done in the same time frame and samples were randomized over 96 well plates before shipment to Novogene. All TransplantLines samples (including newly added 151 healthy controls in the current revised manuscript) were sent to Novogene in one shipment. However, the 10,000 samples from the DMP were sent in different batches. No significant differences were observed between batches. As we point out in response #3.1 data from the DMP was only used to calculate the distance to general population controls in the current study. We performed the same analysis on controls from the TransplantLines project and observed the same significant associations between distance to controls and HRQoL and other objective physical and mental health related phenotypes in the current study (**Figure 4**). Therefore, we are confident that the observed associations are robust and not due to any potential batch effects.

Comment 3.3: In the statistical analysis, a number of covariates were used, reflecting the importance of controlling for potential confounds where possible. However, it is also common to see such analyses presented both with and without covariates, for comparison against other studies where covariates are not included or where different covariates are used, and for transparency so that the reader can understand the “raw” or direct relationships between variables of interest. I would suggest that this could be included as a supplement.

Response 3.3: We agree with the reviewer and accordingly and have now added such an analysis to the supplements. This new analysis involves three models: (1) no confounders; (2) microbiome related confounders (i.e. stool water content, antibiotics, laxatives, proton pump inhibitors); and (3) all identified confounders (i.e. age, sex, BMI, eGFR, stool water content, antibiotics, antihypertensive treatment, diabetes, dialysis, laxative, proton pump inhibitors). Please, also note comment #1.2 of reviewer #1. Incorporation of these models have been made in **lines 223-232, 241-246 and 286-290** of the results section **and Supplementary Tables 7, 9 and 11**.

Comment 3.4: Some of the approaches to the microbiome analysis were rather unusual and perhaps need further explanation. For example, why did the authors decide to look at correlations with PC1? Was this based on previously conducted studies or a theoretical rationale? This is a potential concern given that, in their sample, PC1 explained only 7% of the variance in the microbiome (which seems substantially smaller than what I have typically seen in the literature). Can the authors offer any potential reasons for the low amount of variance explained and provide some discussion on how meaningful correlations are between this subset of microbiome data and the other measures?

Response 3.4:

When we divided HRQoL into quantiles, a gradient appeared along PC1 ranging from Q1 (low score) to Q4 (high score). This is the main reason to why put extra attention on PC1 (that said, we still also analyze PC2-PC5). What is more striking is that we find that it is the same health-disease associated microbial species reported by the Dutch Microbiome Project (DMP) that underlies this gradient; we find that species loading onto its left side (Figure 1AB) are associated with general health in the DMP, and species loading onto its right side (Figure AB) are associated with disease in the DMP (Figure 1E). While 7.5% of variation explained may appear on the lower side as compared to the DMP (PCo1 explains 14.7%) and FINRISK (PCo1 explains 15.3%) population studies, it is important to remember that our study population represents a more homogenous, disease, population compared to a general population. Furthermore, other large population studies, such as DMP and FINRISK have used a different methodology, i.e. Principal Coordinate Analysis together with the Bray-Curtis dissimilarity. This is different from PCA together with centered log-ratio transformed relative abundances (i.e. the Aitchison distance). The latter is becoming the new norm in microbiome research because it is the gold standard when working with so-called compositional (i.e. proportional) data. However, just for comparison, when we apply PCoA and Bray-Curtis dissimilarity on our data, PCo1 explains 11.3% of variation.

Table 4 Variance explained for the first 10 principal components

Principal component (PC)	Variance
PC1	7.513
PC2	4.633
PC3	3.589
PC4	2.751
PC5	2.225
PC6	1.856
PC7	1.816
PC8	1.6
PC9	1.537
PC10	1.474

Comment 3.5: I am also unfamiliar with elastic-net regularization and it is probably worth explaining the advantage of this approach over more commonly used approaches (e.g. HUMAnN or MaAsLin).

Response 3.5: Elastic Net regression is a type of linear regression that combines the advantages of both L1 (Lasso) and L2 (Ridge) regularization methods.⁸ The main advantage of elastic-net regression over linear regression is that it can handle multicollinearity (i.e., high correlations between predictor variables) and can handle situations where the number of predictor variables is larger than the number of observations. Specifically, elastic-net regression includes a penalty term that combines both L1 and L2 regularization. The L1 regularization can perform feature selection by shrinking the coefficients of irrelevant predictor variables to zero, while the L2 regularization can prevent overfitting by shrinking the coefficients of all predictor variables towards zero. Overall, elastic-net regression can provide a more robust and accurate model compared to linear regression when dealing with high-dimensional data with correlated predictor variables.

As MaAsLin is a wrapper function for linear regression, the advantages that is gained from using Elastic Net is a better handling of overfitting (i.e. having too many predictors for the size of your data set) and the bias introduced by multicollinearity (i.e. high correlation between bacterial abundances).⁹⁻¹¹ We have added a section to the methods section explaining our rationale behind choosing elastic-net regression in **lines 193-200 and lines 516-530**. Furthermore, we included a linear model per species into our manuscript as well to make the study more interpretable with commonly used approaches in microbiome studies as well, see **response 3.3**.

Comment 3.6: It was not clear to me why quartiles were used to analyze the HRQoL data, rather than correlations. Can the authors provide some justification for this?

Response 3.6: We have analyzed the relationship between HRQoL and the gut microbiome using correlation analysis. Correlation analysis for principal components and HRQoL data using spearman correlation as is reported in **Supplementary Table 3**. We used quartiles to be able to depict HRQoL in an interpretable way in the ordination plot (**Main Figure 1A and 1B**). For association analysis with principal components, diversity metrics and distance to general population controls we report both spearman correlation as a Mann-Whitney U-test between Q1 and Q4. We chose to do this as it tests a slightly different hypothesis. Correlation analysis is straightforward to interpret while discretizing into quartiles, the PCS and MCS splits the data into the lowest and highest 25% of the cohort for the particular scores. We think this

provides an additional level of interpretation. We hope to increase clarification of these analyses in our current revisions in **lines 121-122 and 509-511**

Comment 3.7: Typically, several measures of alpha diversity are analyzed, reflecting the different aspects of diversity that they measure. Since the Shannon index incorporates evenness, it is probably worthwhile to examine a measure of richness as well.

Response 3.7: We have included richness of bacterial species and the Simpson index into our diversity analysis. Accordingly, we have revised **line 172** of the results section and have updated **Supplementary figure 3 (Figure 6 of rebuttal)**.

Figure 6, Supplementary figure 3 of the main manuscript Diversity index and HRQoL. (A, C and E) Violin plots depict the Shannon diversity, richness of bacterial species and the Simpson index for different quartiles of the physical component scale. (B, D and F) Violin plots depict the Shannon diversity, richness of bacterial species and the Simpson index for different quartiles of the mental component scale. (G) Heatmap depicting significant (white boxes indicate a P-value>0.05) spearman correlations of the Shannon diversity index with HRQoL features from the SF-36-questionnaire.

Comment 3.8: For the beta diversity plots in **Figures 1A & B**, it seems unusual to represent the groups by their centroid (at least, I am assuming that's what the bold dots represent) rather than ellipses to illustrate the overlap between groups. I am also a little confused about the analysis of beta diversity. Was PERMANOVA carried out to determine group differences between the quartiles based on the overall data, or were data analyzed only for each principal component separately? Again, this seems quite unusual and further explanation would be informative.

Response 3.8: We agree with the reviewer that we could have included ellipses and have now done so (**Figure 7 and main Figure 1**). The bold dots indeed represent centroids and we have now added this to the figure legend in **lines 179-180**. We assessed the associations between HRQoL and principal components one to five using spearman correlation on the original HRQoL-scores, i.e. not on quartiles (**Supplementary Table 3**). We have added this to **lines 130 and 160-161**. PERMANOVA analysis was also performed for HRQoL scores (**Supplementary Table 4**) and this was performed on the complete Aitchison distance matrix as we have described in **lines 124-127** of the results section. We have expanded the methods section in **lines 503-514** to better explain the analysis we have performed.

Figure 7, Main figure 1 of manuscript. Disease-associated bacterial species in the gut microbiome of KTR are associated with HRQoL. (A & B) Principal component analysis on the clr-transformed species reflecting the Aitchison distances between KTR. The physical component score **(A)** and the mental component score **(B)** are divided into quartiles (KTR were divided into quartiles (Q1, Q2, Q3, Q4) based on their PCS and MCS with Q1 containing the lowest HRQoL scores and Q4 the highest HRQoL scores). The large dots represent centroids per group and the dashed circles represent 95% confidence ellipses. **(C & D)** Correlation plot between the physical component score **(C)**, the mental component score **(D)** and the distance to general population controls. This dysbiosis score was calculated previously by calculating the Aitchison distance from KTR with 1183 age-, sex- and BMI-matched general population controls¹³. **(E)** Heatmap depicting significant correlations between

species that have previously been associated with disease vs. no disease in the Dutch microbiome project (DMP)¹⁹. Species that are associated with no disease in the DMP (green squares) were consistently, positively and significantly associated with principal component 1 (i.e. higher HRQoL) in our study while the opposite effect was observed for species that were associated with disease in the DMP.

Comment 3.9: In general, there is no mention of how the authors accounted for multiple comparisons in the methods of the analyses (including how FDR was calculated).

Response 3.9: We have added: “We accounted for multiple testing using Benjami-Hochberg correction and an FDR<0.05 was considered as significant.” in **lines 529-530** of the method section.

Comment 3.10: Given that the GBMs were developed to test for potential gut-brain links, it seems surprising that there were many more GBMs related to physical HRQoL than mental HRQoL (in fact, if I interpreted the data correctly, there was only 1 GBM that uniquely associated with mental health?). This is perhaps worth considering in the discussion.

Response 3.10: We agree with the reviewer regarding this rather surprising result. However, in the study by Valles-Colomer et al. two gut brain modules (GBM) were associated with the mental score (MCS), while only isovaleric acid synthesis II was uniquely associated with the MCS.² Many of the gut brain modules contain general functions that play a vital role for general health like short-chain fatty acid metabolism while also influencing the gut brain axis.²⁹ Furthermore, the MCS is strongly correlated with the PCS ($r = 0.65$, $P=1.78 \times 10^{-64}$) and this might be a reason why we only observe 1 uniquely associated GBM with MCS. We have added this to the discussion section in **lines 373-377**.

Comment 3.11: In the discussion the authors focus on butyrate, yet there was no mention of butyrate pathways in the GBM analysis. Were there any significant findings for butyrate in the predictive functional analysis? If not, can the authors reconcile this with their hypotheses on butyrate-producing bacteria?

Response 3.11: In all of our data layers we found butyrate related results. *Faecalibacterium prausnitzii*, *Roseburia hominis*, *Alistipes putredinis*, *Eubacterium hallii*, *Roseburia intestinalis* and *Roseburia inulinivorans* are all butyrate producing bacteria that

were decreased in the gut microbiome of KTR with lower HRQoL.³⁰ We also found MetaCyc pathway 5676 (Acetyl CoA fermentation to butanoate) and gut brain module 053 (butyrate synthesis) to be negatively associated with HRQoL. Therefore, we devoted lines 357-363 of the discussion to butyrate producing species.

Comment 3.12: Overall, the effect sizes seem relatively small, which should be acknowledged when considering whether the microbiome will be a valuable treatment target as proposed.

Response 3.12: We agree with the reviewer that most of the effect sizes are moderate to low and added a statement to the discussion section in **lines 400-401**.

Minor points:

Comment 3.13: For the HRQoL measures, it would be good to know (expected) averages for the general population or standardized bands of functioning if they exist, to allow readers to interpret the averages for this specific population of KTR.

Response 3.13: In the primary analyses, we calculated the physical and mental component scores without using data of a reference population. However, we agree with the reviewer that it is insightful to compare HRQoL scores in our study with the general population. To accommodate the comment of the reviewer, and also to accommodate comment #1.0 and #1.6 of reviewer #1, we also mention the MCS and PCS scores that are standardized based on means, standard deviations and factor score coefficients of the general population of the United States in the revised version of the manuscript in **lines 111-119**.³³ We found that the mean standardized PCS score of our KTR population was 44.5 (SD 10.7), and the mean standardized MCS score was 52.8 (SD 8.6). As a score of 50 is regarded as the mean HRQoL-score of the general population, this implies that the HRQoL of our sample of KTR is on average lower in PCS and higher MCS than the general US population. In total 303 (60%) of KTR had a score < 50 on the PCS and 137 (27%) of KTR had a score < 50 on the MCS. We described the interpretation of these results also in the manuscript. Because the purpose of our study is to investigate the potential link between the microbiome and HRQoL, and not to compare HRQoL of KTR with other populations for which the standardized method is preferred, we performed our primary analyses with the unstandardized method.

Comment 3.14: It should be noted as a limitation that mental health was assessed only through self-report.

Response 3.14: We have added this in **lines 384-385** of the limitation section.

Comment 3.15: Be careful of language that may be interpreted as causal or directional (e.g. line 121: HRQoL “explained’ interindividual variation in the gut microbiome)

Response 3.15: We have changed this statement in **line 135**.

Comment 3.16: Line 342: typo “known to covariate with”

Response 3.16: We changed “covariate” to “be associated” in **line 399**.

Comment 3.17: The recommendation that probiotics could potentially benefit HRQoL seems a bit general – what species might be good candidates to test in this regard, or are there metabolites that could be used as supplements instead?

Response 3.17: We agree with the reviewer and updated **lines 407-410** with the recommendation for a probiotics containing butyrate producing species. We mention this specifically after the limitations section to encourage the reader to consider the suggestion within the limitations of the current study.

Comment 3.18: It would be helpful if page/line references were included in the STORMS checklist

Response 3.18: We have included page and line numbers in the STORMS checklist.

Comment 3.19: It would be helpful if acronyms could be expanded in the supplementary tables (or at least include a key in the index page)

Response 3.19: We have reworked our supplementary tables and removed all acronyms.

References

1. Eisenga, M. F. *et al.* Rationale and design of TransplantLines: a prospective cohort study and biobank of solid organ transplant recipients. *BMJ Open* **8**, e024502 (2018).
2. Valles-Colomer, M. *et al.* The neuroactive potential of the human gut microbiota in quality of life and depression. *Nat Microbiol* **4**, 623–632 (2019).
3. Swarte, J. C. *et al.* Gut microbiome dysbiosis is associated with increased mortality after solid organ transplantation. *Sci. Transl. Med.* **14**, eabn7566 (2022).
4. Gacesa, R. *et al.* Environmental factors shaping the gut microbiome in a Dutch population. *Nature* **604**, 732–739 (2022).
5. Boekhorst, J. *et al.* Stool energy density is positively correlated to intestinal transit time and related to microbial enterotypes. *Microbiome* **10**, 223 (2022).
6. Matsuda, K. *et al.* Direct measurement of stool consistency by texture analyzer and calculation of reference value in Belgian general population. *Sci. Rep.* **11**, 2400 (2021).
7. Douwes, R. M. *et al.* Discrepancy between self-perceived mycophenolic acid-associated diarrhea and stool water content after kidney transplantation. *Clin. Transplant.* **35**, e14321 (2021).
8. Ogutu, J. O., Schulz-Streeck, T. & Piepho, H.-P. Genomic selection using regularized linear regression models: ridge regression, lasso, elastic net and their extensions. *BMC Proc.* **6 Suppl 2**, S10 (2012).
9. Gloor, G. B., Macklaim, J. M., Pawlowsky-Glahn, V. & Egozcue, J. J. Microbiome Datasets Are Compositional: And This Is Not Optional. *Front. Microbiol.* **8**, 2224 (2017).
10. Susin, A., Wang, Y., Lê Cao, K.-A. & Calle, M. L. Variable selection in microbiome compositional data analysis. *NAR Genom Bioinform* **2**, lqaa029 (2020).
11. Hinton, A. L. & Mucha, P. J. A Simultaneous Feature Selection and Compositional Association Test for Detecting Sparse Associations in High-Dimensional Metagenomic Data. *Front. Microbiol.* **13**, 837396 (2022).
12. Blanco-Míguez, A. *et al.* Extending and improving metagenomic taxonomic profiling with uncharacterized species using MetaPhlAn 4. *Nat. Biotechnol.* (2023) doi:10.1038/s41587-023-01688-w.
13. Lins, L. & Carvalho, F. M. SF-36 total score as a single measure of health-related quality of life: Scoping review. *SAGE Open Med* **4**, 2050312116671725 (2016).
14. Knobbe, T. J. *et al.* Airflow Limitation, Fatigue, and Health-Related Quality of Life in Kidney Transplant Recipients. *Clin. J. Am. Soc. Nephrol.* **16**, 1686–1694 (2021).
15. Barnett, C. T., Vanicek, N. & Polman, R. C. J. Temporal adaptations in generic and population-specific quality of life and falls efficacy in men with recent lower-limb amputations. *J. Rehabil. Res. Dev.* **50**, 437–448 (2013).
16. Pekmezović, T. *et al.* Quality of life in patients with progressive supranuclear palsy: one-

- year follow-up. *J. Neurol.* **262**, 2042–2048 (2015).
17. Md Yusop, N. B., Yoke Mun, C., Shariff, Z. M. & Beng Huat, C. Factors associated with quality of life among hemodialysis patients in Malaysia. *PLoS One* **8**, e84152 (2013).
 18. Ware, J. E., Keller, S. D. & Kosinski, M. *SF-36: Physical and Mental Health Summary Scales : a User's Manual.* (1994).
 19. Kanehisa, M., Sato, Y., Kawashima, M., Furumichi, M. & Tanabe, M. KEGG as a reference resource for gene and protein annotation. *Nucleic Acids Res.* **44**, D457–62 (2016).
 20. Vajdi, M. & Farhangi, M. A. A systematic review of the association between dietary patterns and health-related quality of life. *Health Qual. Life Outcomes* **18**, 337 (2020).
 21. Singh, R. K. *et al.* Influence of diet on the gut microbiome and implications for human health. *J. Transl. Med.* **15**, 73 (2017).
 22. Zhernakova, A. *et al.* Population-based metagenomics analysis reveals markers for gut microbiome composition and diversity. *Science* **352**, 565–569 (2016).
 23. Hariharan, S., Israni, A. K. & Danovitch, G. Long-term survival after kidney transplantation. *N. Engl. J. Med.* **385**, 729–743 (2021).
 24. Wang, Y. *et al.* Mapping health-related quality of life after kidney transplantation by group comparisons: a systematic review. *Nephrol. Dial. Transplant* **36**, 2327–2339 (2021).
 25. Salosensaari, A. *et al.* Taxonomic signatures of cause-specific mortality risk in human gut microbiome. *Nat. Commun.* **12**, 2671 (2021).
 26. Aitchison, J. On criteria for measures of compositional difference. *Math. Geol.* **24**, 365–379 (1992).
 27. Martino, C. *et al.* A novel sparse compositional technique reveals microbial perturbations. *mSystems* 4: e00016-19. Preprint at (2019).
 28. Swarte, J. C. *et al.* Characteristics and Dysbiosis of the Gut Microbiome in Renal Transplant Recipients. *J. Clin. Med. Res.* **9**, (2020).
 29. Xiong, R.-G. *et al.* Health Benefits and Side Effects of Short-Chain Fatty Acids. *Foods* **11**, (2022).
 30. Lee, J. R. *et al.* Butyrate-producing gut bacteria and viral infections in kidney transplant recipients: A pilot study. *Transpl. Infect. Dis.* **21**, e13180 (2019).
 31. Vandeputte, D. *et al.* Quantitative microbiome profiling links gut community variation to microbial load. *Nature* **551**, 507–511 (2017).
 32. Galazzo, G. *et al.* How to Count Our Microbes? The Effect of Different Quantitative Microbiome Profiling Approaches. *Front. Cell. Infect. Microbiol.* **10**, 403 (2020).
 33. Ware, J. E., Keller, S. D. & Kosinski, M. *SF-36: Physical and Mental Health Summary Scales : a User's Manual.* (1994).

REVIEWERS' COMMENTS

Reviewer #1 (Remarks to the Author):

The revised manuscript has been improved (including the inclusion of new data and analyses that validate their conclusions) and overall the authors carefully addressed my comments.

I just have some minor comments left:

- I am ok with Response 1.2, but I'd appreciate the authors discussed the results of the new analysis and compared it to the previous one instead of just pointing to Table S7.
- Response 1.3: although I agree this requires some computational effort, I still think it would be best if the authors updated their methods to at least the 2021 versions of metaphlan and humann.
- "This could potentially be attributed to the strong correlation between the PCS and MCS". Please report the correlation between these scores quantitatively.
- Response 1.6: why United States population means were used instead of the Dutch population norms?

Reviewer #2 (Remarks to the Author):

The concerns raised were adequately addressed in this version of the manuscript. I have no additional comments at this point.

Reviewer comments

Reviewer #1 (Remarks to the Author):

The revised manuscript has been improved (including the inclusion of new data and analyses that validate their conclusions) and overall the authors carefully addressed my comments.

Comment 1.0: I am ok with Response 1.2, but I'd appreciate the authors discussed the results of the new analysis and compared it to the previous one instead of just pointing to Table S7.

Response 1.0: We thank the reviewer for the careful reconsideration of our revised work. We have now included discussion regarding the comparison of the elastic net model and per species linear model. In **lines 235-245** we now describe several species that were included in the elastic net model and individually significantly associated with HRQoL features. However, we also stress that there are a lot of differences in the results of the two approaches; for example, some species that were included in the elastic net model were not individually significantly associated with HRQoL features. We discuss that this is most likely due to the difference in the statistical methodology between the two approaches (e.g. elastic net incorporates L1 and L2 regularization). We also included a section in the discussion section in **lines 404-411** discussing the strength and limitations of elastic-net modeling vs. simple linear regression.

Comment 1.1: Response 1.3: although I agree this requires some computational effort, I still think it would be best if the authors updated their methods to at least the 2021 versions of metaphlan and humann.

Response 1.1: Updating the versions of MetaPhlAn and HUMAnN to at least the 2021 versions will not take some computational effort but rather a large amount of computational resources and time. Moreover, it will result in a complete overhaul of our analyses and manuscript. In total, 507 fecal samples from kidney transplant recipients and 8,802 fecal samples from the general population were used in the current study. Updating to MetaPhlAn3 and HUMAnN3 for these 9,309 samples requires substantial computational resources and processing time. We expect that our main findings and results will not change significantly given the incremental change between e.g. MetaPhlAn2 and MetaPhlAn3 compared to a change to MetaPhlAn4. Nevertheless, the results of using a different version will yield different values which also means a redo of all of our analyses, updating all figures and all (supplementary) tables, while the main messages of the current manuscript will not change.

Importantly, one of the main goals of the current study was to compare our results with two studies that were performed in the general population - the study of Valles-Colomer (Nat Microbiol. 2019;4(4):623-632) which relates to the gut brain modules and HRQoL, and our previous study regarding the gut microbiome of the general population in the Netherlands (Gacesa, Nature. 2022 604(7907):732-739), the same population as used in the current study. Both of these studies used MetaPhlAn2 and therefore, we used the same version. Updating our analysis to MetaPhlAn3 would make comparing our study to the results of these two studies difficult and less valid. Given these arguments we have not updated the versions of MetaPhlAn and HUMAnN.

Comment 1.2: "This could potentially be attributed to the strong correlation between the PCS and MCS". Please report the correlation between these scores quantitatively.

Response 1.2: We now report the correlation between the physical component scale and the mental component scale in **lines 111-112**.

Comment 1.3: Response 1.6: why United States population means were used instead of the Dutch population norms?

Response 1.3: We chose to compare with the United States population means for international comparison. However, we now also added the Dutch population means in **lines 115-116**.

Reviewer #2 (Remarks to the Author):

The concerns raised were adequately addressed in this version of the manuscript. I have no additional comments at this point.